# Positive and negative incentive contrasts lead to relative value perception in ants

Stephanie Wendt[1]*, Kim S Strunk[2], Jürgen Heinze[3], Andreas Roider[4], Tomer J Czaczkes[1]

[1]Animal Comparative Economics Laboratory, Institute of Zoology and Evolutionary Biology, University of Regensburg, Regensburg, Germany; [2]School of Business, Economics and Information Systems, Chair of Management, People and Information, University of Passau, Passau, Germany; [3]Institute of Zoology and Evolutionary Biology, University of Regensburg, Regensburg, Germany; [4]Department of Economics, University of Regensburg, Regensburg, Germany

**Abstract** Humans usually assess things not according to their absolute value, but relative to reference points – a main tenant of Prospect Theory. For example, people rate a new salary relative to previous salaries and salaries of their peers, rather than absolute income. We demonstrate a similar effect in an insect: ants expecting to find low-quality food showed higher acceptance of medium-quality food than ants expecting medium quality, and vice versa for high expectations. Further experiments demonstrate that these contrast effects arise from cognitive rather than mere sensory or pre-cognitive perceptual causes. Social information gained inside the nest can also serve as a reference point: the quality of food received from other ants affected the perceived value of food found later. Value judgement is a key element in decision making, and thus relative value perception strongly influences which option is chosen and ultimately how all animals make decisions.

DOI: https://doi.org/10.7554/eLife.45450.001

*For correspondence: wendtstephanie@outlook.de

Competing interests: The authors declare that no competing interests exist.

## Introduction

We all compare options when making both large and small decisions, ranging from career choices to the choices of an evening's entertainment. Understanding how options are compared has thus been central to the study of behavioural economics. Theories explaining the mechanisms by which options are compared and decisions are made have a long tradition (*Vlaev et al., 2011*), with Expected Utility Theory (EUT) being the most widely used theory in economic models (*Mankiw, 2011*; *von Neumann and Morgenstern, 1944*). EUT suggests that decisions are made by evaluating and comparing the expected utility from each option. A rational decision maker then chooses the option resulting in the best end state: the option providing the greatest utility (*von Neumann and Morgenstern, 1944*).

However, over the past decades economic research on human decision making has started to shift away from a view of (absolute) utility maximisation towards more nuanced notions of relative utility, such as reference-dependent evaluations. *Kahneman and Tversky (1979)* made a major contribution to this shift by introducing Prospect Theory, suggesting that decision making is not based on absolute outcomes, but rather on relative perceptions of gains and losses. In contrast to EUT, the utility attributed to options being evaluated is determined relative to a reference point, such as the status quo or former experience (*Vlaev et al., 2011*; *Kahneman and Tversky, 1979*; *Parducci, 1984*; *Tversky and Kahneman, 1992*; *Ungemach et al., 2011*). Various examples of relative value perception have been described. For example, satisfaction gained from income is perceived not absolutely, but relative to the income of others in the social reference group – such as one's colleagues

**eLife digest** We make many decisions every day, often by comparing options and choosing the one with the greatest profit. But how much we value something often does not depend solely on our needs. Instead, this value may depend on our expectations or other arbitrary reference points. For example, how satisfied you are with your income might depend on how much your colleagues or friends earn.

Animals, including insects, also make decisions when feeding, choosing a partner, or finding a nesting site. Sometimes animals behave in ways that look like disappointment. For example, monkeys may reject a cucumber as a reward if they have seen another monkey get a grape for completing the same task. But it is hard to tell if this behavior reflects a value judgment.

To investigate whether insects evaluate their options against their expectations, Wendt et al. offered black garden ants sugar water over multiple trials. Some ants grew to expect low quality sugar water (containing little sugar); some expected medium quality; and others expected high quality sugar water (containing a high concentration of sugar). Ants that expected to find low quality sugar water were more likely to accept medium quality options than ants that expected the medium quality sugar water. Similarly, ants that expected high quality sugar water were less likely to accept lower quality sugar water. Further experiments confirmed that the ants were not using physical cues such as satiation to guide their behavior.

Furthermore, Wendt et al. found that ants that returned to the nest after foraging passed on information that altered the expectations of the next group of foragers about nearby food. This suggests that the value that ants place on food sources depends both on individual experiences and on information gained from others.

Studies of decision making in humans can be difficult to perform and interpret, because volunteers may try to second-guess what the experimenters want to find, and culture and education may also influence choices. Studying ants instead could help to avoid these pitfalls, as the results presented by Wendt et al. suggest they make decisions in similar ways to humans. Future work building on these findings could also help researchers to predict how insects behave, particularly in rapidly changing environments.

DOI: https://doi.org/10.7554/eLife.45450.002

(*Boyce et al., 2010*). Overall, Prospect Theory has enriched our understanding of human decision making by conceptualising it as more nuanced and less rational than previously assumed (*Tversky and Kahneman, 1974*; *Tversky and Kahneman, 1981*).

The concept of malleable value perception is not just relevant to humans. Value judgments in animals are also influenced by factors apparently independent of the absolute value of options. For example, capuchin monkeys refuse otherwise acceptable pay (cucumber) in exchanges with a human experimenter if they had witnessed a conspecific obtain a more attractive reward (grape) for equal effort (*Brosnan and De Waal, 2003*). Rats, starlings, and ants, like humans, place greater value on things they work harder for (*Aw et al., 2011*; *Czaczkes et al., 2018a*; *Lydall et al., 2010*), and starlings, fish and locusts demonstrate state-dependent learning, wherein they show a preference for options experienced when they were in a poor condition (*Aw et al., 2009*; *Pompilio et al., 2006*; *Schuck-Paim et al., 2004*). Roces and Núñez (*Roces, 1993*; *Roces and Núñez, 1993*) aimed to show that in leaf cutting ants perceived value can be influenced by other ants. Ants recruited to higher quality food sources ran faster, deposited more pheromone, but cut smaller leaf fragments, even if the food source the recruits find is replaced by a standardised food source (*Roces, 1993*; *Roces and Núñez, 1993*). However, in these experiments, the absolute value and nature of the reference remains unclear, and indeed pheromone presence may have caused the observed behaviours without influencing the ants' expectations or value perception at all.

*Healey and Pratt (2008)* showed that colonies of the house-hunting ant species *Temnothorax curvispinosus* move into a nest of mediocre quality faster when they were previously housed in a high-quality nest compared to colonies which were previously housed in a poor-quality nest (*Healey and Pratt, 2008*). In contrast, *Stroeymeyt et al. (2011)* showed that colonies of *Temnothorax albipennis* developed an aversion towards mediocre-quality nests available in their

environment when they were housed in a high-quality nest, whereas colonies housed in a low-quality nest did not, and thus show an experience-dependent flexibility in nest choice (*Stroeymeyt et al., 2011*). However, critically missing from the existing works is a systematic description of value judgment relative to a reference point.

'Value distortion by comparison' effects have been studied for decades using the successive contrasts paradigm, in which animals are trained to a quality or quantity of reward which is then suddenly increased (positive incentive contrast) or decreased (negative incentive contrast) (*Bentosela et al., 2009*; *Bitterman, 1976*; *Couvillon and Bitterman, 1984*; *Crespi, 1942*; *Flaherty, 1982*; *Flaherty, 1999*; *Mustaca et al., 2000*; *Weinstein, 1970a*). Many mammals, including apes, monkeys, rats, and dogs (*Brosnan and De Waal, 2003*; *Bentosela et al., 2009*; *Crespi, 1942*; *Flaherty, 1999*; *Mustaca et al., 2000*; *Pellegrini and Mustaca, 2000*; *Weinstein, 1970b*) have been shown to respond to successive negative contrast by disrupting their behaviour compared to control animals which had not experienced a change in reward. The animals display behaviour akin to disappointment – slower running speeds to a reward (*Bower, 1961*), depressed licking behaviour (*Flaherty et al., 1985*; *Vogel et al., 1968*), or reward rejection (*Tinklepaugh, 1928*).

Contrast effects were also successfully described in invertebrates (*Bitterman, 1976*; *Couvillon and Bitterman, 1984*; *Richter and Waddington, 1993*). *Bitterman (1976)* found negative incentive contrast effects in honey bees which were trained to a high-quality feeder and then received a downshift to a lower quality feeder. In contrast, bees which experienced an upshift in feeder quality did not show any feeding interruptions (*Bitterman, 1976*; *Couvillon and Bitterman, 1984*). While negative successive contrast effects – akin to disappointment – have been well described in animals, positive successive contrast effects – akin to elation – have often proved elusive (*Bower, 1961*; *Black, 1968*; *Capaldi and Lynch, 1967*; *Dunham, 1968*; *Papini et al., 2001*). There are several factors which may prevent positive contrast effects from being detected. Firstly, ceiling effects may occur when the performance of animals receiving a large reward is at or near a physical limit. The absence of positive contrast effects may then not be due to the absence of perceived positive contrast, but rather due to an artefact of experimental design (*Bower, 1961*; *Campbell et al., 1970*). Secondly, neophobia counteracts positive contrast effects: animals may be reluctant to eat a novel food – even if the food is of higher quality than normal (*Flaherty, 1999*; *Oberhauser and Czaczkes, 2018*). Finally, generalisation decrement may prevent stronger responses to positive contrast. Generalisation decrement occurs when animals are trained under one set of stimuli and then tested under another. The strength of the tested response may decrease with increasing differences between the training and testing stimuli (*Kimble, 1961*), which may then result in weaker positive contrast effects following a reward shift. Thus, the reward change itself may lead to a decrease in responding just as would any other change in context, such as a change in the brightness of the runway or scent of the food (*Oberhauser and Czaczkes, 2018*; *Capaldi, 1978*; *Premack and Hillix, 1962*).

Even though positive contrast effects proved to be hard to demonstrate in laboratory experiments, there are good theoretical reasons for expecting both positive and negative contrast effects to evolve (*McNamara et al., 2013*). Incentive contrasts have also been demonstrated for rewards other than food. Females become more (or less) likely to accept a mate of given quality if they have prior experience of better (or worse) mates. Such mate quality contrast effects are reported in both vertebrates (*Collins, 1995*) and invertebrates (*Dukas, 2005*; *Reid and Stamps, 1997*; *Wagner et al., 2001*).

In this study, we investigate positive and negative contrast effects using the successive contrasts paradigm, and, in addition to demonstrating positive and negative contrast effects, define the first relative value curve in an invertebrate; the ant *Lasius niger*. We conduct a critical control experiment to rule out physiological or psychophysical effects which may lead to the same pattern (see experiment 2) and thus provide strong evidence for a purely cognitive relative value effect in a non-human animal. Furthermore, we demonstrate that information flowing into the nest can influence value perception in outgoing foragers. This suggests that food sources are not only valued based on individual experiences, but also based on social information gained inside the nest. The perceived value of a food source influences social information dissemination, by affecting the strength of pheromone trails which then lead further ants to the food source. Thus, the way in which value is judged is likely to strongly affect the foraging mechanics of a whole colony.

# Materials and methods

## Study animals

Eight stock colonies of the black garden ant *Lasius niger* were collected on the University of Regensburg campus. The colonies were kept in 30 × 30 × 10 cm foraging boxes with a layer of plaster covering the bottom. Each box contained a circular plaster nest box (14 cm diameter, 2 cm height). The colonies were queenless with around 1000–2000 workers and small amounts of brood. Queenless colonies still forage and lay pheromone trails and are frequently used in foraging experiments (*Devigne and Detrain, 2002*; *Dussutour et al., 2004*). The colonies were fed with *ad libitum* 0.5M sucrose solution and received *Drosophila* fruit flies once a week. Water was available *ad libitum.*

One sub-colony of 500 individuals was formed from each stock colony, and these eight fixed-size sub-colonies were used for our experiments. Sub-colonies were maintained identically to the stock colonies, but did not receive any *Drosophila* fruit flies to prevent brood production, and were starved 4 days prior to the experiments in order to achieve a uniform and high motivation for foraging (*Mailleux et al., 2006*; *Josens and Roces, 2000*). During starvation, water was available *ad libitum.* Any ants which died or were removed from the sub-colonies were replaced with ants from the original stock colonies.

## General setup, ant selection, and monitoring

The general setup used for all of our three experiments was identical and consisted of a 20 × 1 cm long paper-covered runway which was connected to the sub-colony's nest box via a 40 cm long drawbridge (*Figure 1A*). A 5 mm diameter drop of sucrose solution (Sigma-Aldrich) was placed on an acetate feeder at the end of the runway (60 cm from the nest). The molarity of the sucrose droplet depended on the experiment, treatment and on the ants' number of visit to the food source.

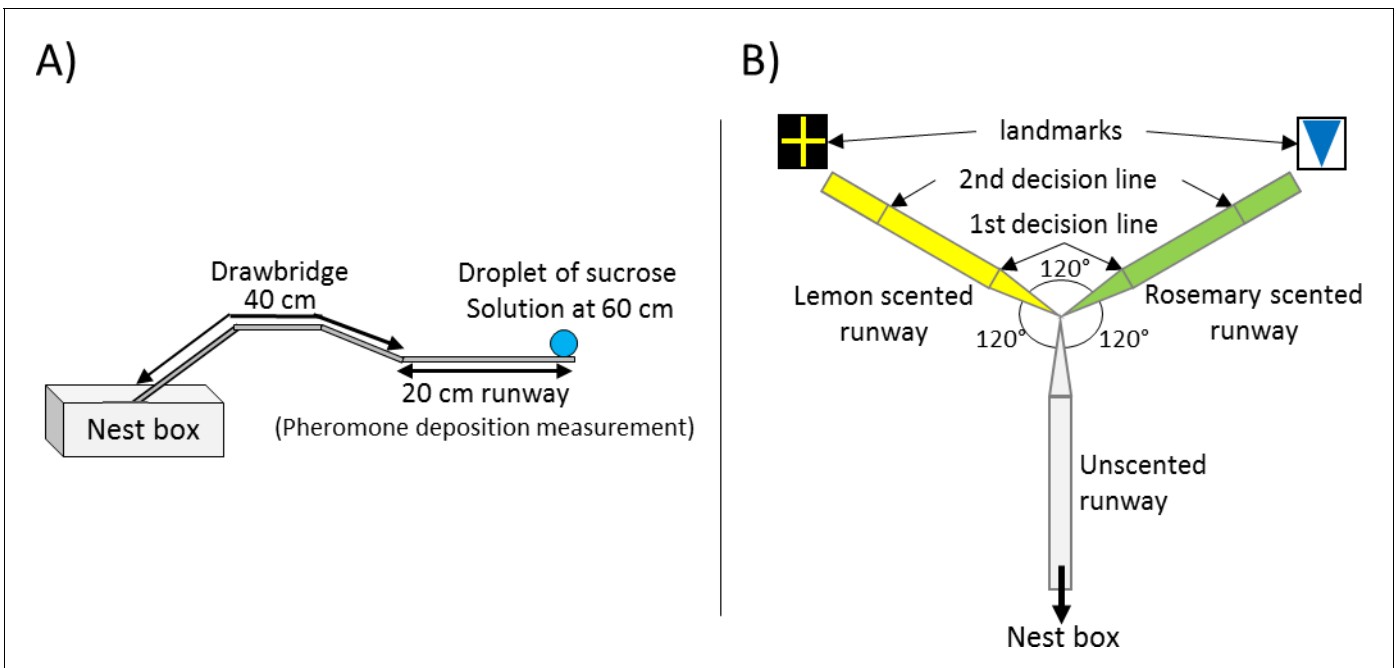

**Figure 1.** Experimental Setups. (**A**) General setup used for all presented experiments. The 20 cm long runway is connected to the nest box via a 40 cm long drawbridge. The droplet of sucrose solution is placed at the end of the runway (60 cm distance to the nest). (**B**) Y-maze used on the 10th visit of experiment 2. All arms were 10 cm long. The arm connected to the nest box was covered with unscented paper overlays while the other two arms were covered with lemon and rosemary scented paper overlays (one scent on each side). Visual cues (landmarks) were placed directly behind the two scented arms. The first decision line was located 2.5 cm from the Y-maze centre and marked the initial decision of an ant while the second decision line was located 7.5 cm from the centre and marked the final decision.
DOI: https://doi.org/10.7554/eLife.45450.003

To begin an experiment, 2–4 ants were allowed onto the runway, and the first ant to reach the feeder was marked with a dot of acrylic paint on its gaster. This procedure may select for the more active foragers, but does not introduce any selection bias between treatments. The marked ant was allowed to drink to repletion at the food source, while all other ants were returned to the nest.

Food acceptance scores as a measure of perceived value were noted for each ant and visit as follows: Full acceptance (1) was scored when the ant remained in contact with the drop from the moment of contact and did not interrupt drinking within 3 s of initial contact (see *Video 1*). Partial acceptance (0.5) was scored if feeding was interrupted within 3 s after the first contact with the food source, but the ant still filled its crop within 10 min (as can be seen by the distention of the abdominal tergites). Ants which interrupt feeding within the first seconds after contacting the food usually show successive feeding interruptions and generally show a rather 'impatient' behaviour compared to ants which show a food acceptance score of 1 (see *Video 2*). Lastly, rejection (0) was scored if the ant refused to feed at the sucrose solution and either returned to the nest immediately or failed to fill its crop within 10 min.

When the ant had filled its crop or decided not to feed at the sucrose droplet, it was allowed to return to the nest. Inside the nest, the ant unloaded its crop to its nestmates and was then allowed back onto the runway for another visit. The drawbridge was now used to selectively allow only the marked ant onto the runway.

In addition to measuring food acceptance, we also measured pheromone deposition. Individual pheromone deposition behaviour correlates with the (perceived) quality of a food source (*Beckers et al., 1993*; *Hangartner, 1970*; *Czaczkes et al., 2015*). Individual ants can adapt the strength of a pheromone trail by either depositing pheromone or not, or varying the intensity of a pheromone trail through number of pheromone depositions (*Beckers et al., 1993*; *Hangartner, 1970*). Pheromone deposition behaviour in *L. niger* is highly stereotypic. To deposit pheromone, an ant briefly interrupts running to bend its gaster and press the tip of the gaster onto the substrate (*Beckers et al., 1992*). This allows the strength of a pheromone trail to be quantified by counting the number of pheromone depositions over the 20 cm runway leading to the feeder. Pheromone depositions were measured each time the ant moved from the food source back to the nest (inward trip), and each time the ant moved from the nest towards the food source (outward trip). Because *L. niger* foragers almost never lay pheromone when they are not aware of a food source (*Beckers et al., 1992*), we did not measure pheromone depositions for the very first outward trip (visit 1). The presence of trail pheromone on a path depresses further pheromone deposition (*Czaczkes et al., 2013*). Thus, each time an ant had passed the 20 cm runway, the paper overlay covering the runway was replaced by a fresh one every time the ant left the runway to feed at the feeder or returned to the nest.

All experimental runs were recorded with a Panasonic DMC-FZ1000 camera to allow for later video analysis. Each tested ant was observed until all experimental runs were finished and then discarded from the colony before switching to the next ant. If an ant did not return before

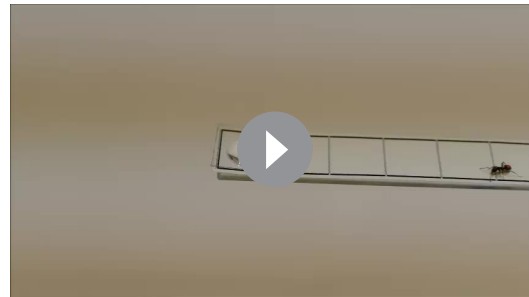

**Video 2.** Ant displaying food acceptance score 0.5. It interrupts feeding within the first seconds of feeding and repeatedly interrupts feeding, but still feeds at the food source (an ant displaying food acceptance score 0 would refuse to feed at the sucrose solution and either return to the nest immediately or fail to fill its crop within 10 min).
DOI: https://doi.org/10.7554/eLife.45450.005

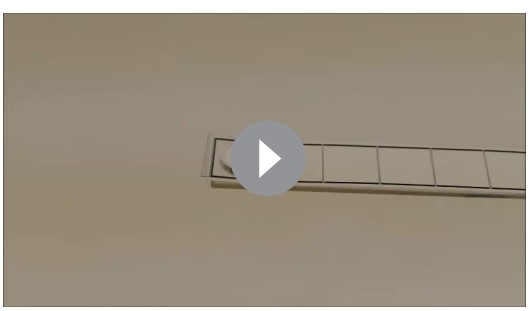

**Video 1.** Ant displaying food acceptance score 1. It shows no food interruptions within the first seconds of feeding.
DOI: https://doi.org/10.7554/eLife.45450.004

finishing all experimental runs, we waited for 15 min, then discarded it from the colony and moved to the next ant.

## Statistical analysis

Statistical analyses were carried out in R v. 3.4.1 (*R Development Core Team, 2016*) using Generalized Linear Mixed Models (GLMMs) in the LME4 package (*Bates et al., 2014*) to analyse pheromone depositions data and Cumulative Link Mixed Models (CLMMs) in the ordinal package (*Christensen, 2015*) to analyse food acceptance scores. CLMMs were used to analyse the acceptance data since we used an ordered factor with three levels (1 = full acceptance, 0.5 = partial acceptance, 0 = rejection).

As multiple ants were tested per colony, colony identity was added as a random effect to each model. GLMMs were tested for fit, dispersion and zero inflation using the DHARMa package (*Hartig, 2017*). The model predictors and interactions were defined a priori, following *Forstmeier and Schielzeth (2011)*. All p-values presented were corrected for multiple testing using the Benjamini–Hochberg method (*Benjamini and Hochberg, 1995*). A total of 1070 ants were tested, with 829 in experiment 1, 73 in experiment 2 and 168 in experiment 3 (*Supplementary file 1*). Sample sizes were set ahead of time by deciding how much time we will invest in data collection (1 day per treatment per colony).

## Food acceptance data

Depending on the experiment, we either used treatment (experiment 1 and 3 = Reference Molarity; experiment 2 = expected molarity triggered by a scented runway and the odours presented on the runway) or an interaction between treatment and visit number, and the odours presented on the runway (training visits of experiment 2) or trophallaxis time (experiment 3) as fixed factors. The interaction between expected molarity and visit number in the training runs of experiment 2 was added, because experience with a food source is likely to affect the behaviour at a food source. The odours presented on the runway were added as fixed factors to test for odour preferences regardless of sucrose molarity. The interaction between trophallaxis time and reference molarity in experiment 3 was added because trophallaxis time may affect food acceptance through crop load and information gained through trophallaxis (for the effects of trophallaxis time on food acceptance see *Figure 5—figure supplement 1*, and Table S4 in *Figure 5—source data 1*). Because individual ants were tested multiple times in experiments 1 and 2, we included AntID nested in colony as a random factor for statistical analyses of the training visits.

We used the following general model formula (this formula varied depending on experiment as described above):

$$FoodAcceptance \sim treatment + (random\ factor : colony)$$

## Pheromone deposition data

As the pheromone deposition data is count data, they were analysed using a GLMM with a Poisson distribution.

Depending on the experiment, we either used treatment (experiment 1 = Reference Molarity; experiment 2 = expected molarity triggered by a scented runway and the odours presented on the runway) or an interaction between treatment and visit number (training visits of experiment 2) as fixed factors. The interaction between expected molarity and visit number in the training runs of experiment 2 was added, because experience with a food source is likely to affect the behaviour at a food source. The odours presented on the runway were added as fixed factors to test for odour preferences regardless of sucrose molarity. Because individual ants were tested multiple times in experiment 2, we included AntID nested in colony as a random factor for statistical analyses of the training visits.

For statistical analysis of experiment 1, we also added a variable indicating if ants deposited more or less pheromone compared to the average to correct for individual strength of pheromone depositions and overdispersion. The variable was calculated as follows:

$$\text{Difference to average} = ((\text{Number Pheromone Depositions } 1^{\text{st}} \text{ visit} - \\ \text{mean number Pheromone Depositions } 1^{\text{st}} \text{ visit}) + \\ (\text{Number Pheromone Depositions } 2^{\text{nd}} \text{ visit} - \text{mean number Pheromone Depositions } 2^{\text{nd}} \text{ visit})) / 2$$

We used the following model formulae in the model:
Experiment 1:

$$\text{NumberPheromoneDepositions} \sim \text{treatment} + \text{Difference to average} + \\ (\text{Difference to average})^2 + (\text{random effects} : \text{colony} / \text{AntID})$$

Experiment 2:

$$\text{NumberPheromoneDepositions} \sim \text{scent associated to molarity} + (\text{random effects} : \text{colony})$$

## Other analyses

The number of drinking interruptions was quantified via video analysis in experiment 2 (see below). This was analysed statistically in a manner identical to number of pheromone depositions.

Trophallaxis time in seconds in experiment three were used in full seconds and treated as count data. We performed a GLMM with Poisson distribution and Reference Molarity as a fixed effect, while colony identity was added as a random factor:

$$\text{TrophallaxisTimeseconds} \sim \text{ReferenceMolarity} + (\text{random effects} : \text{colony})$$

## Experiment 1 – Defining a relative value perception curve

The aim of this experiment was to test whether *Lasius niger* ants value a given absolute sucrose solution concentration relative to a reference point or based on its absolute value. We used a range of twelve molarities as reference points in order to describe a value curve. To exclude effects of the researcher's expectations on the data, the data for this experiment were collected blind to treatment (*Holman et al., 2015*).

## Experiment 1 - Methods

Ants made two initial training visits to a feeder at the end of a runway in order to set their reference point (*Figure 1A*). The quality of the sucrose solution was varied between ants, with each ant receiving the same quality twice successively. Twelve different molarities were used: 0.1, 0.2, 0.3, 0.4, 0.5, 0.6, 0.7, 0.8, 0.9, 1, 1.5 or 2M (also referred to as pre-shift solution or reference point). *Lasius niger* workers learn the quality of a feeder within two visits (*Wendt and Czaczkes, 2017*). On the third visit (test visit), the food source was replaced by a 0.5M sucrose solution droplet for all ants (also referred to as post-shift solution). Thus, ants trained to qualities < 0.5M experienced a positive successive contrast, ants trained to > 0.5M experienced a negative successive contrast, and the ants trained to 0.5M constituted the control (no contrast). 97% of ants successfully finished the training procedure and participated in the test visit (third visit).

## Experiment 1 - Results

Ants seemed to value sucrose solution droplets relative to their reference point (*Figure 2—figure supplement 1*). In the training visits, acceptance scores increased significantly with increasing molarity of the reference quality (CLMM: estimate = 1.97, z = 9.65, p<0.001, *Figure 2*). However, in the test (contrast) visit, acceptance scores decreased significantly with increasing molarity of the reference quality (CLMM: estimate = −2.59, z = −13.57, p<0.001, *Figure 2*). Ants which were trained to the lowest molarity (0.1M: p<0.001) showed significantly higher acceptance of 0.5M sucrose than control ants, while ants trained to high molarities (1.5M: p<0.001, 2M: p<0.001) showed lower acceptance of 0.5M than the control group (see Table S1 in *Figure 2—source data 1* for all pairwise comparisons).

A similar pattern was found for pheromone deposition behaviour on the way back to the nest (*Figure 3*). In the training visits, number of pheromone depositions increased significantly with increasing molarity of the reference solution (GLMM: estimate = 0.86, z = 13.87, p<0.001). By contrast, on the test visit pheromone depositions decreased significantly with increasing molarity of the reference solution (GLMM: estimate = −0.82, z = −9.75, p<0.001, *Figure 3*). Ants which deposited

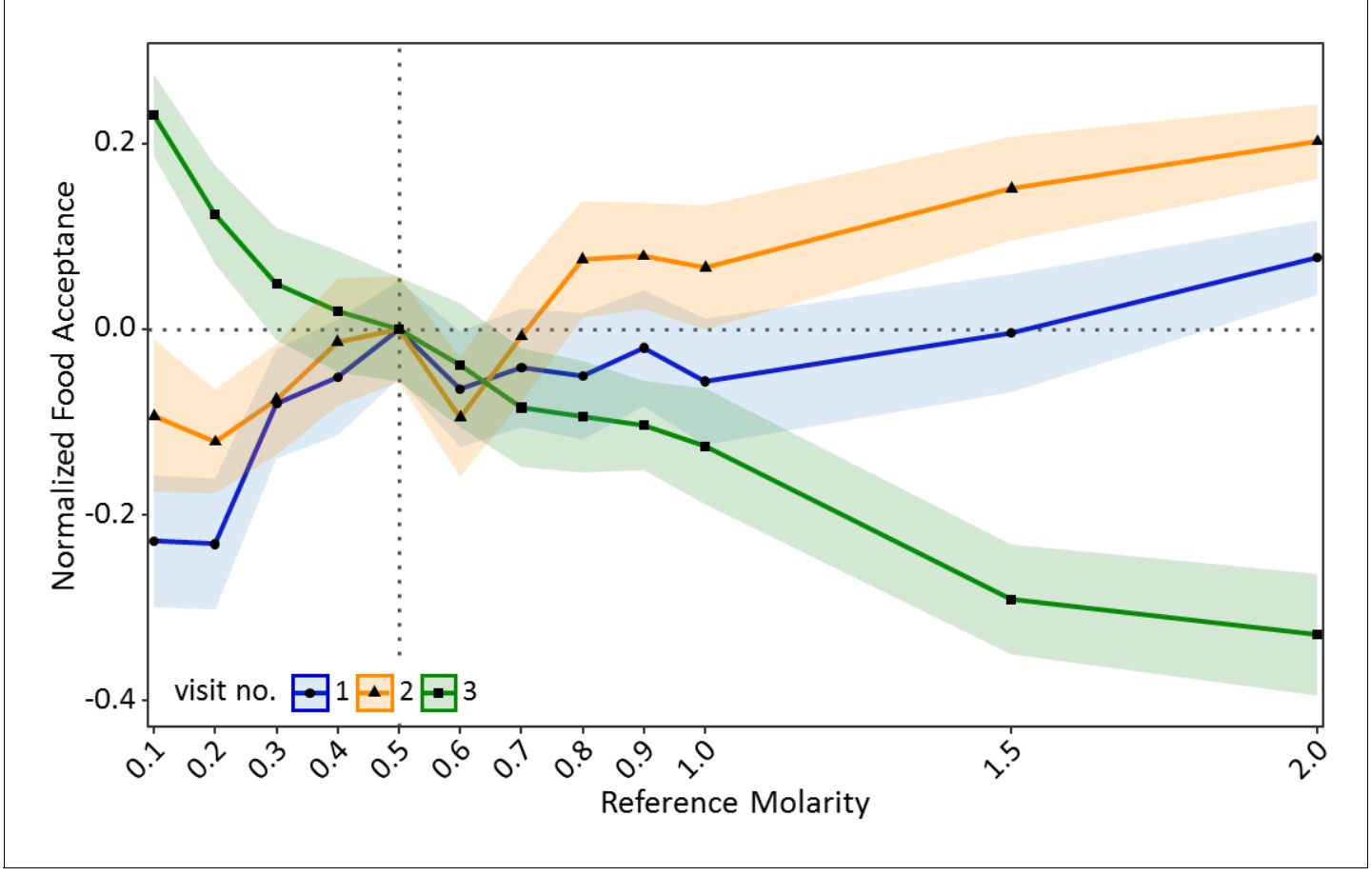

**Figure 2.** Food acceptance shown in experiment one for the two training visits (visit 1 and 2) in which ants received one of 12 molarities and the test visit (3) in which all ants received 0.5M (sample sizes: 0.1M: 57; 0.2M: 80; 0.3M: 76; 0.4M: 66; 0.5M: 77; 0.6M: 65; 0.7M: 73; 0.8M: 66; 0.9M: 72; 1M: 55; 1.5M: 72; 2M: 70). Shown are the mean food acceptance (points) and the 95% confidence intervals (coloured ribbons) for each reference molarity and visit. Data was normalised to show the mean food acceptance of the control group (received 0.5M on each visit) at 0 for all three visits. For a non-normalised graph of the data see *Figure 2—figure supplement 1*.

DOI: https://doi.org/10.7554/eLife.45450.006

The following source data and figure supplement are available for figure 2:

**Source data 1.** Experiment 1 – Defining a relative value perception curve; Data Analysis of the Food Acceptance scores for training visits 1 and 2 and test visit 3.
DOI: https://doi.org/10.7554/eLife.45450.008

**Figure supplement 1.** Food acceptance shown in experiment one for the two training visits (visit 1 and 2) in which ants received one of 12 molarities (Reference Molarity) and the test visit (3) in which all ants received 0.5M.

DOI: https://doi.org/10.7554/eLife.45450.007

more pheromone during the training visits generally deposited more pheromone on the test visit compared to ants which deposited less pheromone during the training visits (GLMM: estimate = 0.16, z = 15.99, p<0.001). Ants which were trained to a low molarity (0.2M: p<0.01) deposited significantly more pheromone in the test visit than control ants, while ants trained to high molarities (1M: p<0.001, 1.5M: p<0.001, 2M: p<0.001) deposited less pheromone than the control group (see Table S2 in *Figure 3—source data 1* for pairwise comparisons).

These results are consistent with relative value perception stemming from the psychological effects of successive contrasts. We could further define a relative value perception curve similar to that described in Prospect Theory, as well as showing positive contrast effects for both food acceptance and number of pheromone depositions.

However, there is another possible explanation for these results: non-random selection of individuals with different acceptance thresholds. Different individuals from the same colony may have

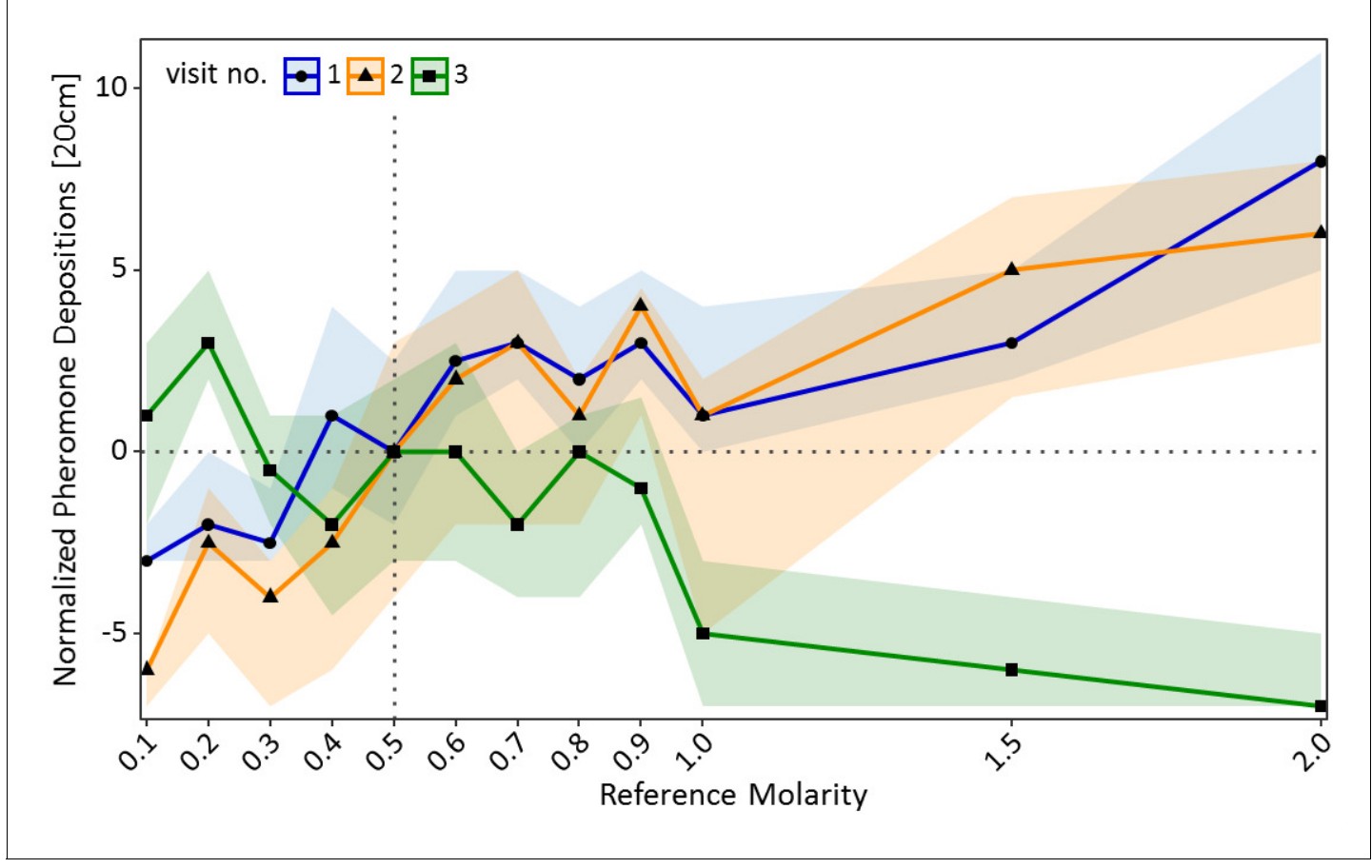

**Figure 3.** Pheromone depositions on the way back to the nest shown in experiment one for the two training visits (visit 1 and 2) in which ants received one of 12 molarities and the test visit (3) in which all ants received 0.5M (sample sizes: 0.1M: 57; 0.2M: 80; 0.3M: 76; 0.4M: 66; 0.5M: 77; 0.6M: 65; 0.7M: 73; 0.8M: 66; 0.9M: 72; 1M: 55; 1.5M: 72; 2M: 70). Shown are the median number of pheromone depositions (points) and the 95% confidence intervals (coloured ribbons) measured on a 20 cm track right behind the food source for each reference molarity and visit. Data was normalised to show the median number of pheromone depositions of the control group (received 0.5M on each visit) at 0 for all three visits. For a non-normalised graph of the data see *Figure 3—figure supplement 1*.

DOI: https://doi.org/10.7554/eLife.45450.009

The following source data and figure supplement are available for figure 3:

**Source data 1.** Experiment 1 – Defining a relative value perception curve; data analysis of the inbound pheromone depositions (to Nest) for training visits 1 and 2 and test visit 3.

DOI: https://doi.org/10.7554/eLife.45450.011

**Figure supplement 1.** Pheromone depositions on the way back to the nest shown in experiment 1 for the two training visits (visit 1 and 2) in which ants received one of 12 molarities (Reference Molarity) and the test visit (3) in which all ants received 0.5M.

DOI: https://doi.org/10.7554/eLife.45450.010

different acceptance thresholds. Animals with lower acceptance thresholds may readily exploit low-quality food sources while animals with higher thresholds may not. When training to lower molarity sucrose, ants with high thresholds may not have completed training, leaving only a non-random subset of ants with low acceptance thresholds at the test phase (*Robinson et al., 2009*). Thresholds may also be influenced by experience, by which animals use the best experienced option as a threshold for accepting a new option or not (*Stroeymeyt et al., 2011*; *Robinson et al., 2011*). However, we can exclude this possibility, as the proportion of ants not completing training was uniformly low and did not vary with treatment (see *Supplementary file 1*).

## Experiment 2 – ruling out alternative explanations using scent training

Alternative hypotheses could also explain the results from experiment one and lead to the same behavioural patterns observed. Five possible 'lower-level' mechanisms must be excluded: sensory satiation, ingested sucrose changing haemolymph-sugar levels, psychophysical sensory contrast effects, the fact that ants may expect pre-shift solutions to return in later visits, and non-random selection of individuals with different food acceptance thresholds in different treatments.

### Sensory satiation

This may occur in ants which were trained to higher molarity food due to the blocking of more sweetness receptors compared to low molarity sucrose. The more sweetness receptors are blocked by a sweet reference solution, the fewer receptors will fire when confronted with a post-shift reward, thus making solutions taste less sweet for ants trained to high-molarity solutions, and sweeter for ants which were trained to low molarities (*Bitterman, 1976*).

### Haemolymph-sugar levels

Ants may not only have stored sucrose solutions in their crop while foraging, but may also have ingested small amounts of it, leading to an increase of haemolymph-sugar levels. Higher blood-sugar levels negatively affect sweetness perception in humans (*Mayer-Gross and Walker, 1946*; *Melanson et al., 1999*), and a similar effect could cause a post-shift solution to taste less sweet to animals trained on high sucrose concentrations.

### Psychophysical sensory contrasts

The contrast effects shown in experiment one could also derive from simple psychophysical mechanisms (*Fechner, 1860*; *Zwislocki, 2009*), and thus arise from sensory perceptual mechanisms rather than higher level cognitive processing of value. Sensory judgements are usually made relative to reference points and through constant comparisons with former stimuli (*Vlaev et al., 2011*; *Helson, 1964*). Thus, identical stimuli may be perceived differently depending on the context they are presented within. The position of the reference point in the range of stimuli may thus bias how the stimulus, and thus the value, of a post-shift reward is perceived (*Zwislocki, 2009*). For example, the sweetness of a sucrose solution may be perceived as much stronger when the reference point to which it is compared is low. Psychophysical sensory contrasts are physiological or low-level cognitive phenomena, found in all animal taxa studied, and even potentially in bacteria (*Akre and Johnsen, 2014*; *Kojadinovic et al., 2013*; *Mesibov et al., 1973*).

### Future expectations

Animals may rationally expect the pre-shift reward to be available in the future again and therefore rationally show lower acceptance towards the post-shift reward, because they are waiting for the pre-shift reward to reoccur.

All these alternative factors would lead to the same behavioural patterns found in experiment one without relative value perception necessarily being present. Experiment two was designed to rule out these alternative explanations.

### Experiment 2 - Methods

To rule out the alternative non-psychological explanations for the contrast effects we described above, we needed to change the expectations of the ants while exposing all ants to identical training regimes. This would provide a reference point for testing relative value perception while keeping sensory saturation, haemolymph-sugar levels, psychophysical effects, future expectations, and ant subsets the same until the testing phase.

Ants were allowed to make eight training visits. The quality of the sucrose solution offered at the end of the runway alternated each visit, always beginning with the low-quality solution. The solutions were scented using either rosemary or lemon essential oils (0.05 µl essential oil per ml sucrose solution, rosemary: *Rosmarinus officinalis*; Lemon: *Citrus limon*, Markl GbR, Grünwald). In half the trials the 1.5M solution was scented with lemon and the 0.25M with rosemary, and vice versa for the other

trials. In addition, to support learning and to allow solution quality anticipation, we also scented the paper overlays covering the runway leading to the feeder. Paper overlays were scented by storing them for at least 1 day in an airtight box containing a droplet of essential oil on filter paper in a pet-ridish. Finally, in addition to odours cuing sucrose molarity, visual cues were also provided. These consisted of printed and laminated pieces of paper (22 × 16.5 cm, displayed in *Figure 1B*) displayed at the end of the runway, directly behind the sucrose droplet.

On the 9th (test) visit, the odour of the runway and the visual cue signified either 1.5M or 0.25M, while the sucrose solution provided was unscented and of intermediate (0.5M) quality. Runway scents in the test visit were varied systematically between ants, but each ant was confronted with only one of the two runway scents coupled with unscented 0.5M sucrose. While the ant fed at the sucrose droplet, the scented runway overlay was replaced with an unscented overlay in order to eliminate possible effects of scent association on pheromone deposition behaviour. Previous work has shown that *L. niger* foragers can form robust expectations of upcoming reward quality based on runway odour after four visits to each odour/quality combination (*Czaczkes et al., 2018b*). Nonetheless, to ensure that learning had taken place, on the 10th visit, we carried out a memory probe. The linear runway was replaced with a Y-maze (*Figure 1B*), with two 10 cm long arms and a 10 cm long stem. The Y-maze stem was covered with an unscented paper overlay while one arm was covered with the 1.5M-associated odour overlay, and the other with the 0.25M-associated odour overlay. The matching visual cues were placed directly behind the relevant Y-maze arms. Trained ants were allowed to walk onto the Y-maze and their arm choice was noted. We used two decision lines to define arm choice – an initial decision line (*Figure 1B*, 2.5 cm after the bifurcation) and a final decision line (7.5 cm after the bifurcation). After testing on the Y-maze, the ants were permanently removed from the colony.

97.2% of ants successfully finished the training procedure and participated in the last test visit.

Additionally to the other measures, on the 9th (test) visit of this experiment we counted the number of food interruptions made by an ant from the moment of first hitting the food source until it had finished feeding at the sucrose droplet. The number of food interruptions are likely to reflect and support the behaviour encoded in food acceptance scores and was thus investigated to give stronger support for the results of this experiment.

## Experiment 2 - Results

During training, ants behaved as expected, showing higher acceptance and pheromone deposition for 1.5M compared to 0.25M on all but the very first visit to 0.25M (Food acceptance: CLMM: estimate = −7.34, z = −8.9, p<0.001; pheromone depositions outward journey: GLMM: estimate = 0.23, z = 2.89, p<0.01; pheromone depositions inward journey: GLMM: estimate = −2.49, z = −19.46, p<0.001, *Figure 4A,C & E*). Furthermore, food acceptance and pheromone depositions both on the outward and inward journeys decreased with increasing experience with the 0.25M feeder and increased with increasing experience with the 1.5M feeder (Food acceptance: CLMM: estimate = −2.84, z = −3.63, p<0.001; pheromone depositions outward journey: GLMM: estimate = −0.94, z = −10.00, p<0.001; pheromone depositions inward journey: GLMM: estimate = −0.53, z = −4.41, p<0.001).

On the outward journey of the 9th (test) visit, ants walking towards the feeder while exposed to 1.5M sucrose-associated cues deposited more pheromone (median = 15, *Figure 4D*) compared to ants exposed to 0.25M-associated cues (median = 2, GLMM: estimate = −1.31, z = −12.94, p<0.001). Moreover, in the learning probe, 87% of ants chose the 1.5M associated arm. This demonstrates that ants formed a robust expectation of food molarity based on the cues learned during training.

Ants exposed to 1.5M-associated cues during the 9th visit showed significantly lower food acceptance towards the unscented 0.5M feeder than ants exposed to 0.25M-associated cues (CLMM: estimate = 1.04, z = 2.049, p<0.05, *Figure 4B*, *Supplementary file 1*). Although ants exposed to high molarity associated cues – presented through scented runways on the way to the food – showed a significantly higher number of pheromone depositions on their return journey than ants confronted with low molarity scent (GLMM: estimate = −1.65, z = −3.03, p<0.01, *Figure 4E & F*), the number of pheromone depositions decreased drastically for both treatments compared to training visits (median 1.5M = 0, median 0.25M = 0, *Figure 4E and F*, *Supplementary file 1*).

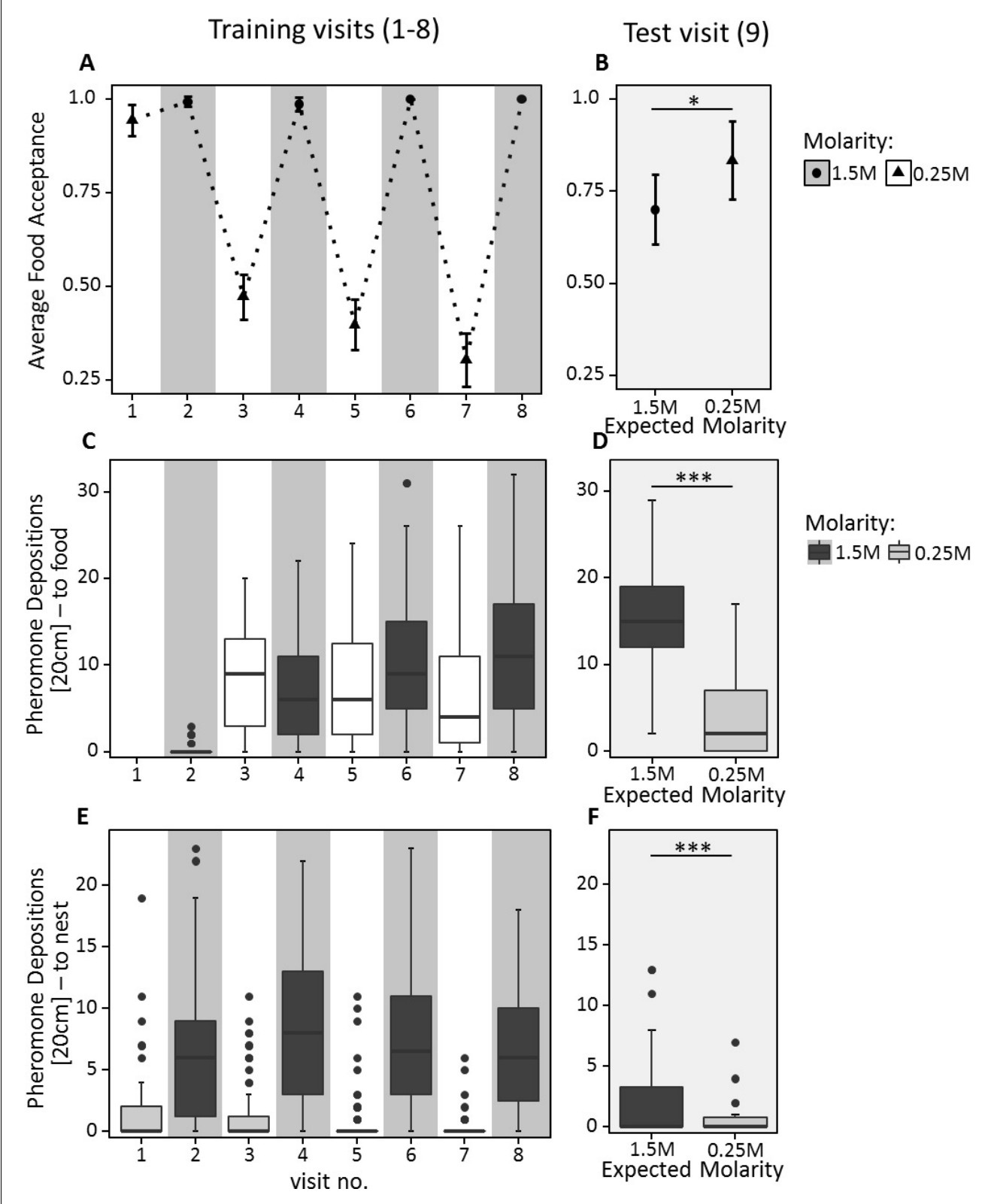

**Figure 4.** Food acceptance (**A** and **B**) and number of pheromone depositions towards the food source (**C** and **D**) and towards the nest (**E** and **F**) in experiment 2. The left panels (**A**, **C**, **E**) show behaviour over the eight training trials, in which ants received 0.25M coupled with one scent and 1.5M coupled with another scent on alternating visits. The right panels (**B**, **D**, **F**) show behaviour on the test visit, in which ants always received unscented 0.5M sucrose solution, but the runway leading towards the food source was impregnated with one of the learned scents, causing ants to expect either a

*Figure 4 continued on next page*

*Figure 4 continued*

high or low reward. 40 ants were induced to expect a high reward, and 32 to expect a low reward. **A** and **B** show the mean food acceptance (points) and the 95% confidence intervals (error bars) for each visit; **C** – **F** show the median number of pheromone depositions on a 20 cm track leading to the food source and the 75%/25% quantiles for each visit.

DOI: https://doi.org/10.7554/eLife.45450.012

The following source data and figure supplements are available for figure 4:

**Source data 1.** Experiment 2 – ruling out alternative explanations using scent training; data analysis of the food acceptance scores for training visits 1 to 8.

DOI: https://doi.org/10.7554/eLife.45450.015

**Source data 2.** Experiment 2 – ruling out alternative explanations using scent training; Data analysis of the food acceptance scores for test visit 9.

DOI: https://doi.org/10.7554/eLife.45450.016

**Source data 3.** Experiment 2 – ruling out alternative explanations using scent training; data analysis of the outbound pheromone depositions (to Food) for training visits 1 to 8.

DOI: https://doi.org/10.7554/eLife.45450.017

**Source data 4.** Experiment 2 – ruling out alternative explanations using scent training; data analysis of the outbound pheromone depositions (to Food) for test visit 9.

DOI: https://doi.org/10.7554/eLife.45450.018

**Source data 5.** Experiment 2 – ruling out alternative explanations using scent training; data analysis of the inbound pheromone depositions (to Nest) for training visits 1 to 8.

DOI: https://doi.org/10.7554/eLife.45450.019

**Source data 6.** Experiment 2 – ruling out alternative explanations using scent training; data analysis of the inbound pheromone depositions (to Nest) for test visit 9.

DOI: https://doi.org/10.7554/eLife.45450.020

**Figure supplement 1.** Number of food interruptions on the last (9th) visit depending on the ant's expectations until the crop was filled.

DOI: https://doi.org/10.7554/eLife.45450.013

**Figure supplement 1—source data 1.** Experiment 2 – ruling out alternative explanations using scent training; data analysis of the drinking interruption behaviour.

DOI: https://doi.org/10.7554/eLife.45450.014

---

Even after controlling for alternative explanations, ants still show contrast effects depending on the quality of the post shift solution. This is in spite of all ants undergoing identical training experiences. The only difference between the groups was the odour of the runway on the 9th (test) visit. It is thus unlikely that sensory saturation, increased haemolymph-sugar levels, simple psychophysical effects or ants expecting pre-shift solutions to return can fully explain the behaviour of the ants in our experiments. All videos were re-analysed by a naive scientific assistant and this blind analysis of the ants behaviour confirmed the stated results (CLMM: estimate = 1.42, z = 2.35, p=0.019), and also found that ants interrupted drinking significantly more often when expecting high rather than low food qualities (GLMM, estimate = 0.36, z = 2.74, p=0.006, see *Figure 4—figure supplement 1—source data 1* and *Figure 4—figure supplement 1*).

Non-random selection of individuals with different acceptance thresholds can also be excluded for the results of this experiment as the proportion of ants not completing training was again uniformly low (see *Supplementary file 1*) and all ants had to taste both low and high molarities in order to complete training.

## Experiment 3 – expectation setting via trophallaxis: the nest as an information hub

Ants receive information about available food sources, such as food odour and palatability, through food exchanges (trophallaxis) inside the nest (*Provecho and Josens, 2009*; *Josens et al., 2016*). An ant beginning a food scouting bout may not have direct information about the quality of the food sources available in the environment, but nonetheless must make a value judgement on their first visit to a food source. The aim of this experiment was to ascertain whether information about sucrose concentrations gained through trophallaxis in the nest affected the perceived value of food sources found outside the nest.

## Experiment 3 - Methods

An ant was allowed to feed at an unscented sucrose solution droplet of either 0.16, 0.5 or 1.5M (also referred to as pre-shift solution or reference point) and return to the nest to unload its crop via trophallaxis. When trophallaxis began, we noted the time spent in trophallaxis with the first trophallactic partner. When trophallaxis stopped, the receiving trophallactic partner (receiver) was gently moved from the nest and placed onto the start of a runway offering unscented 0.5M sucrose solution at the end (also referred to as post-shift solution). As the receiver fed, we noted its food acceptance.

## Experiment 3 - Results

Acceptance scores of receivers towards 0.5M decreased with increasing molarity of the sucrose solution received through food exchanges inside the nest (CLMM: estimate = −0.57, z = −3.07, p<0.01). The interaction of reference molarity and trophallaxis time significantly predicted acceptance (CLMM: estimate = −0.48, z = −2.33, p=0.02, *Figure 5*) and longer trophallaxis times led to lower food acceptance in ants as well (CLMM: estimate = −0.70, z = −3.62, p<0.001). Ants which received 0.16M inside the nest showed significantly higher acceptance of 0.5M sucrose than ants which received 1.5M (p<0.01, see Table S3 in *Figure 5—source data 1* for pairwise comparisons). The time spent in trophallaxis with the receiver increased significantly with increasing molarity (GLMM: estimate = 0.13, z = 4.79, p<0.001, see *Figure 5—source data 1*).

Ants valued a standard quality food source relative to the molarity which they received from a returning forager inside the nest. This suggests that information about the quality of a food source received through trophallactic interactions inside the nest can be used by naive foragers when evaluating new food sources outside the nest. Thus, the nest serves as an information hub in which information about available food sources can be gathered, processed, and disseminated.

## Discussion

The introduction of Prospect Theory (*Kahneman and Tversky, 1979*) contributed to a major shift in economic research by suggesting that humans do not perceive value in absolute terms, but relative to reference points. Here, we demonstrate parallel findings in an insect. To the best of our knowledge, we provide the first detailed description of relative value perception in an invertebrate based on individual experience, but also induced by social information. Furthermore, we demonstrate the elusive positive contrast effects in ants which were trained to low molarities (*Figure 2 and 3*).

Similar results in house-hunting ants were explained by a simple threshold rule (*Stroeymeyt et al., 2011*; *Robinson et al., 2009*; *Robinson et al., 2011*) which suggests that individuals have different acceptance thresholds and ants with lower thresholds accept lower quality options. The higher the quality of

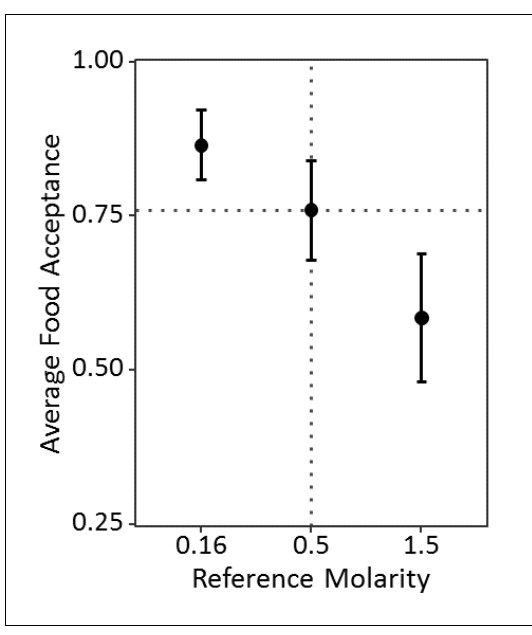

**Figure 5.** Food acceptance shown in experiment 3 for the receivers which received either 0.16, 0.5 or 1.5M through trophallaxis in the nest and then found 0.5M at the end of the runway (sample sizes: 0.16M 63; 0.5M: 52; 1.5M: 53). Shown are the mean food acceptance (points) and the 95% confidence intervals (error bars) for each reference molarity.
DOI: https://doi.org/10.7554/eLife.45450.021

The following source data and figure supplement are available for figure 5:

**Source data 1.** Experiment 3 – expectation setting via trophallaxis: the nest as an information hub; data analysis of the food acceptance scores and trophallaxis time in seconds.
DOI: https://doi.org/10.7554/eLife.45450.023

**Figure supplement 1.** Food acceptance scores dependent on the trophallaxis time [sec] of receiving foragers for all three reference molarities (each plot represents data for 1 of 3 reference molarities).
DOI: https://doi.org/10.7554/eLife.45450.022

the option, the more often it exceeds the acceptance threshold of individual ants, and thus the option is accepted more readily. This could have potentially affected our results in experiment 1, as we would expect fewer individuals to accept very low reference points. Ants which did not accept the low-quality sucrose would thus not be tested. Therefore, at low reference points, we would only select individuals with very low acceptance thresholds, while no threshold selection would occur at high reference points. When confronting ants with medium-quality food after training, the differently selected acceptance thresholds may lead to the same pattern as we observed. However, 97% of all ants finished both the training and the test phases and no higher proportion of cancelled training can be seen at lower reference molarities (see *Supplementary file 1*). It is thus unlikely that a simple threshold rule leads to the results shown in experiment 1 (*Figure 2 and 3*).

While a second major prediction of Prospect Theory, that 'losses loom larger than gains' (*Tversky and Kahneman, 1992*), is not supported by the data of our main experiment, it is also not ruled out. We believe ants do overemphasise losses, but, due to limitations in the experimental design and physiological limitations of the animals, we cannot make strong claims about this (*Collins, 1995*; *Dukas, 2005*; *Reid and Stamps, 1997*). The lack of strong evidence for losses being overemphasised may stem from the psychophysics of our study system: a basic tenant of psychophysics is that the Just Noticeable Difference (JNDs) between two stimuli is a function of the relative difference between the stimuli (*Fechner, 1860*; *Zwislocki, 2009*; *Stevens, 1957*). Thus, ants shifted from 0.1M to medium (0.5M) quality experience a fivefold increase in molarity, while those downshifted from 0.9M to 0.5M experience less than a twofold decrease, although the absolute change was of the same magnitude. This would predict larger shift-changes, in terms of absolute molarity change, for gains than for losses. Indeed, the fact that this is also not seen may imply that losses are indeed – relatively speaking – looming larger than gains for the ants. Finally, it must be kept in mind that acceptance scores are unlikely to be linear, and that pheromone deposition behaviour shows large variation (*Beckers et al., 1992*), making it difficult to use either of these factors to quantitatively test for over- and undervaluation of gains and losses.

The results of experiment 2 allow us to exclude all but a cognitive relative value effect (*Figure 4*). This cognitive effect is subjectively very familiar to humans, and its presence in an invertebrate is at first glance surprising. However, insects have been shown to display many cognitive traits in parallel with humans (*Brosnan and De Waal, 2003*; *Aw et al., 2011*; *Premack and Hillix, 1962*; *Dukas, 2005*; *Reid and Stamps, 1997*), and contrast effects are likely selected for (*McNamara et al., 2013*).

The smaller effect size in experiment two is presumably driven either by the exclusion of the additional driving factors (see experiment two description), or the additional complications involved in an extensive training regime, or both. Specifically, the expectations leading to contrast effects in experiment two were driven by differential learning of odour-quality associations, rather than a simple one-component memory of food quality as may have been the case in experiment 1. This may have weakened the observed effect.

Another possible explanation for smaller effect sizes may be that in experiment 2 ants had access to two reference points (0.25M and 1.5M) to use for value judgement of the medium-quality food in the control experiment, while in experiment 1 they only had one reference. Thus, while the odour cue may have overemphasised the role of the associated quality as reference, the competing reference quality may have been acting as a second reference. Additional reference points are likely to affect the scale post-shift rewards are compared to (*Zwislocki, 2009*). This possibility is supported by the acceptance data collected during training in experiment 2. On the first training visit, all ants encountered low quality food and showed a high food acceptance towards the feeder (*Figure 4A*). However, as soon as ants had experienced a high-quality sucrose solution, the previously acceptable low-quality food became unattractive, and food acceptance scores decreased from a mean of 0.99 to 0.39. This strongly suggests that the ants were valuing the training solutions in relation to each other, and may therefore have used both reference points to judge the value of an unscented medium-quality food source. It is possible that the ants may have calculated an average from both reference points, and used the average as comparison to judge the value of the post-shift reward (*Flaherty, 1999*), as shown in rats (*Peters and McHose, 1974*). However, the fact that medium quality elicited different food acceptance scores depending on the runway scent makes it unlikely that this would be the only factor affecting acceptance scores.

Lastly, masking effects may also explain the smaller contrast effects of experiment 2 compared to experiment 1: learning theory suggests that neutral cues associated to positive stimuli will elicit positive responses even when no reward is given and vice versa (*Rescorla and Wagner, 1972*). Therefore, since ants were confronted with the scent associated to high-quality food, food acceptance may have been affected by the scent itself, leading to an elevated food acceptance compared to ants tested in experiment 1 which did not receive a positive cue, but only medium-quality food.

The reduced pheromone deposition seen in the final return in experiment 2 may be due to the change in environment (scented runways to unscented runways) causing a disruption in recruitment behaviour, perhaps due to generalisation decrement (*Kimble, 1961*; *Capaldi, 1978*) or neophobia (*Barnett, 1958*; *Johnson, 2000*; *Mitchell, 1976*; *Pliner and Loewen, 1997*). Furthermore, since only the scented paper overlays were replaced by unscented ones, but not the runways themselves, it is possible that small portions of the odours were still present, driving the ants to deposit pheromone according to the remaining odours, with higher deposition rates for the high-quality associated odour. In a separate experiment, such pheromone deposition directly related to quality-associated odours on runways was clearly demonstrated (*Wendt and Czaczkes, 2019*). This would explain why pheromone depositions were higher for ants returning to the nest from a high molarity scent than in ants returning from a low molarity scent.

Information about sucrose concentrations gained through trophallactic interactions inside the nest can affect the way newly discovered food sources are valued outside the nest (*Figure 5*), as well as providing other information (*Provecho and Josens, 2009*; *Josens et al., 2016*; *LeBoeuf et al., 2016*). By taking into account information gained inside the nest, recruited workers are able to evaluate newly discovered food sources in relation to other food sources available in the environment. Ants could thus forego food sources which are of lower quality than the average available food sources (*Wendt and Czaczkes, 2017*). Even though higher trophallaxis times led to lower acceptance scores and trophallaxis times were higher at high reference molarities, this does not necessarily imply that ants ingested more sucrose at higher references and were thus less hungry or motivated. Higher sucrose solutions are more viscose and thus ants take longer to ingest the same amount of sucrose compared to low molarities (*Josens et al., 1998*). If, however, more sucrose solution was transferred between the returning forager and the recruit at longer trophallaxis times, it is likely that information input increases and food acceptance decreases. The longer the trophallaxis time, the more the recruit can fill its crop through trophallaxis and therefore the food acceptance may decrease, because the recruit is less starved than an ant which showed a short trophallaxis time. However, even if more food was transferred, the food acceptance scores are a measure of the first assessment of ants at a food source, not the ingested volume. Thus, while some ants may have had less space in their crop left, this may not necessarily affect the food acceptance score, while it is very likely to affect ingested volume after trophallaxis. Additionally, if longer trophallaxis times lead to more ingested sucrose solution, it is also more likely that a higher amount of information about the past food quality is transferred. Thus, more transferred food during trophallaxis may have led to better informed ants reaching the post-shift solution and thus stronger contrast effects. Since the data shows clear effects of both trophallaxis time and reference solution on the food acceptance of 0.5M sucrose, longer trophallaxis times cannot be the only factor driving the contrast effects found in this experiment (see *Figure 5—source data 1* and *Figure 5—figure supplement 1*). Even at high trophallaxis times, ants with a 0.16M reference showed no low food acceptance scores, unlike ants with high reference solution after long trophallaxis times.

Ultimately, we see the nest serving as an information hub, in which information about currently available food sources can be collected, synthesised, and fed back to outgoing foragers. Relative value perception can therefore be expected to have strong effects not only on the individual behaviour of animals, but also on the collective behaviour of insect colonies. For example, colonies of house-hunting ants developed an aversion towards mediocre nests when housed in high-quality nests, but not when they were housed in low-quality nests. Such mediocre nests are then avoided when colonies have to find a new nest site while newly discovered mediocre nests are readily accepted (*Stroeymeyt et al., 2011*; *Robinson et al., 2011*). However, while in house-hunting the reference resource is directly experienced by scouts only, we demonstrate that information brought back to the nest can set a reference point for ants which have not directly experienced the resource in situ.

A broad range of behaviours relevant to behavioural economics have been described in invertebrates (*Czaczkes et al., 2018a*; *Pompilio et al., 2006*; *Wendt and Czaczkes, 2017*; *Czaczkes et al., 2018b*; *Cheng et al., 2002*). We propose that invertebrates make attractive models for a broader understanding of behavioural economics in humans. The benefits of an interdisciplinary approach will likely flow both ways. Using animal models allows researchers to avoid pitfalls associated with studies on humans, such as cultural and educational differences (*Carter and Irons, 1991*; *Guiso et al., 2006*), second-guessing of experimenters, and non-relevant reward sizes (*Levitt and List, 2007*) as well as relaxing ethical concerns. The game-like designs of many economic experiments are highly artificial and the incentive magnitudes that can be provided are limited (*Kahneman and Tversky, 1979*; *Levitt and List, 2007*). While there has been much progress in field studies on humans to clearly measure causal relationships (*Harrison and List, 2004*), the usefulness of these new techniques is constrained by the range of questions and settings to which they can be applied. Hence, while behavioural studies on invertebrates also have their limitations (for example, in that inducing expectations is more of a challenge), they can be easily designed to be ecologically meaningful, and offer rewards which are in line with the real-life budgets under which the animals operate. Finally, due to human complexity, building economic models which accurately predict human behaviour is challenging. Insect economic behaviours are demonstrably similar to that of humans, but likely simpler. We therefore propose that economic models to predict invertebrate decision making may be a complementary step on the way to predicting human behaviour.

There is a well-developed tradition of integrating economics and biology (*Aw et al., 2011*; *Czaczkes et al., 2018a*; *Lydall et al., 2010*; *Aw et al., 2009*; *Wendt and Czaczkes, 2017*; *Cheng et al., 2002*; *Evans and Westergaard, 2006*). Here we provide a systematic description of value judgment relative to a reference point in ants, define a relative value curve as described in Prospect Theory, and provide some of the first strong evidence for a purely cognitive element to relative value judgement. Reference points can not only be set by individual experiences but also through social information such as pheromone trails or through trophallactic contacts inside the nest. We feel a critical mass of evidence is now available to consider comparative behavioural economics as a relevant discipline for both biologists and economists.

## Acknowledgements

We thank Flavio Roces for helpful comments on this work, Florian Hartig for advice concerning statistical analysis of our data, and Nathalie Stroeymeyt, Stephen Pratt, and an anonymous reviewer for comments on an earlier version of this manuscript. We also thank the DFG (Deutsche Forschungsgemeinschaft) which funded SW and TJC with an Emmy Noether grant to TJC, grant number CZ 237/1–1.

## Additional information

### Funding

| Funder | Grant reference number | Author |
| --- | --- | --- |
| Deutsche Forschungsgemeinschaft | Emmy-Noether-Grant (Grant number CZ 237/1-1) | Stephanie Wendt Tomer J Czaczkes |

The funders had no role in study design, data collection and interpretation, or the decision to submit the work for publication.

### Author contributions

Stephanie Wendt, Conceptualization, Data curation, Formal analysis, Validation, Investigation, Methodology, Writing—original draft, Writing—review and editing; Kim S Strunk, Andreas Roider, Validation, Writing—review and editing; Jürgen Heinze, Resources, Validation, Writing—review and editing; Tomer J Czaczkes, Conceptualization, Resources, Formal analysis, Supervision, Funding acquisition, Validation, Methodology, Writing—original draft, Project administration, Writing—review and editing

## Author ORCIDs

Stephanie Wendt (iD) https://orcid.org/0000-0002-8950-2845
Kim S Strunk (iD) https://orcid.org/0000-0002-2896-814X
Tomer J Czaczkes (iD) https://orcid.org/0000-0002-1350-4975

## Ethics

Animal experimentation: All animal treatment guidelines applicable to ants under German law have been followed.

## Decision letter and Author response

Decision letter https://doi.org/10.7554/eLife.45450.029
Author response https://doi.org/10.7554/eLife.45450.030

# Additional files

## Supplementary files

• Supplementary file 1. Sample sizes, mean food acceptance and median pheromone depositions (inward and outward journeys) for the test visits of each experiment and treatment.
DOI: https://doi.org/10.7554/eLife.45450.024

• Transparent reporting form
DOI: https://doi.org/10.7554/eLife.45450.025

## Data availability

Raw data has been deposited on Dryad, https://doi.org/10.5061/dryad.77q6s30. Videos of food acceptance scores and other supplementary data such as statistical analyses have been uploaded as online supplementary files.

The following dataset was generated:

| Author(s) | Year | Dataset title | Dataset URL | Database and Identifier |
|---|---|---|---|---|
| Wendt S, Strunk KS, Heinze JPD, Roider APD, Czaczkes TJD | 2019 | Data from: Positive and negative incentive contrasts lead to relative value perception in ants | https://dx.doi.org/10.5061/dryad.77q6s30 | Dryad Digital Repository, 10.5061/dryad.77q6s30 |

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
