## [Decision Letter]

[Editors’ note: a previous version of this study was rejected after peer review, but the authors submitted for reconsideration. The first decision letter after peer review is shown below.]

Thank you for submitting your work entitled "Positive and negative incentive contrasts lead to relative value perception in ants" for consideration by *eLife*. Your article has been reviewed by three peer reviewers, and the evaluation has been overseen by a Reviewing Editor and a Senior Editor. The following individuals involved in review of your submission have agreed to reveal their identity: Nathalie Stroeymeyt (Reviewer #1); Stephen Pratt (Reviewer #3).

Our decision has been reached after consultation between the reviewers. Based on these discussions and the individual reviews below, we regret to inform you that your work will not be considered further for publication in *eLife*.

While the three reviews found value to the study, they raised concerns about the paper lacking important information (all three reviewers), possible alternative explanations for the results (reviewers 1 and 2) and lack of references to previously published work. Given the extent of the revisions necessary, we are returning the paper to you; however, we would be willing to consider a new submission with a revised manuscript if you can adequately address the comments from the reviewers.

Reviewer #1:

This manuscript presents experimental evidence that the likelihood of an ant worker accepting food of a given quality is influenced by its prior experience with food of higher or lower quality. They authors argue that the underlying mechanism is a cognitive process akin to relative value perception in humans, which depends directly on expectations, with individuals being more likely to reject a food source of a given quality if they had high expectations, and vice versa. Overall, I find the study well-thought-out and carried out to a high scientific standard, and the manuscript unusually well-written. If true, the finding that expectations influence value judgment in ants is both novel and exciting, as prior studies which have highlighted the effect of prior experience on decision-making in insects have not attempted to study the underlying cognitive mechanisms. However, I think that an alternative hypothesis could also explain the experimental results presented in the manuscript, and I would like the authors to provide additional information so this alternative explanation may be (hopefully) ruled out. Furthermore, some crucial information should be moved from the Supplementary material to the main text to help the readers understand the experimental design and judge the validity of the results without having to constantly refer to the SI. Finally, the results should also be discussed in light of some relevant studies on ant decision-making which are currently not cited in the paper. If all these points can be addressed, I believe there would be a strong argument for publication in *eLife*.

1) The results from experiment 1 and 2 could be explained by an alternative hypothesis in which each ant has a fixed, experience-independent threshold for food acceptance. (This alternative hypothesis is inspired from the following work on house-hunting in ants: Robinson et al., 2009, Robinson et al., 2011, Stroeymeyt et al., 2011).

In experiment 1, the authors explain that they allow 4 ants on the drawbridge, and wait until the first ant finds the food. That ant becomes the focal experimental ant and her behavior is monitored during that visit plus two subsequent trips, then is discarded from the colony. However, no information is given regarding whether any ant had to be discarded from the experiment because they did not perform three trips. One could imagine a scenario in which ants are more likely to lose motivation to forage when the food source they found is of lower quality than their internal, fixed acceptance threshold. Under that scenario, when trained with a food source of very high quality, all or most ants would keep motivation to forage, whereas when trained with a food source of low quality, only ants with a low acceptance threshold would keep motivation to forage. Thus during the third, test visit, the populations of tested foragers would not be equivalent between treatments but would show a sampling bias: in the first treatment (high quality training food), tested foragers would present a broad variation of acceptance threshold and would thus be more likely to reject an average-quality food source during the third visit, whereas in the second treatment (low quality training food), tested foragers would only include ants with low acceptance threshold, which would thus all accept an average-quality food source. In order to reject this scenario, it is necessary that the authors present data on the proportion of focal ants that did not do three trips. If that proportion does not increase with decreasing quality of the training food, then the alternative hypothesis I presented can be safely rejected, which would provide more support for the authors' interpretation.

Similarly, in experiment 2, it is possible that ants with a high acceptance threshold make a U-turn and return to the nest when reaching the scented portion of the bridge when it carries the smell associated with the low-quality food, while all ants may cross the bridge and reach the food when the bridge carries the smell associated with the high-quality food. Under that scenario, the two test populations would again not be equivalent, and the results would be explained by self-organized sampling bias rather than by a cognitive expectation effect. I would thus like the author to include data on the proportion of ants that performed U-turns when reaching the scented portion of the bridge in the two treatments. If there is no difference between the two treatments, then the alternative hypothesis can again be safely rejected and this would provide more support for the authors' interpretation.

Finally, I am not convinced that the results from experiment 3 cannot be solely explained by the different crop loads of the receiver ants in the three treatments: as the receiver ants receive more food from donor ant when the original food source is of higher quality, and as they were not given any opportunity to unload before being confronted with the test food, then one would predict them to drink less during the test (and so to have a lower food acceptance score) simply because they have less space in their crop. I am not sure that the statistical analyses presented by the authors are enough to demonstrate an effect beyond this simple mechanical effect. Instead of Figure S6.3, can the author present a scatterplot of mean acceptance score as a function of trophallaxis time for each of the three treatments and elaborate on the interpretation of the statistical analyses presented in the Supplementary material? Given that the main effect 'Reference Molarity' is non-significant and that the sign of the estimate (+) is opposite to that expected based on the author's hypothesis (higher reference molarity should result in lower food acceptance), it seems that the conclusions drawn from experiment 3 hold may not be supported by the data. Please elaborate and modify the text if relevant.

2) When reading the main text, I had trouble evaluating the solidity of the authors' claim because of the lack of key information which is only provided in the Supplementary material. In particular, please provide the following information in the main text:

- definition of the food acceptance score

- information that each ant was uniquely identified owing to the design of the experiment (each experimental ant monitored fully then discarded from the colony before switching to the next ant)

- more detail on the alternative scenarios at the beginning of experiment 2

- more detail on the experimental ant selection procedure, to show that your experimental design does not introduce any a priori selection bias between treatments

3) Previous work on house-hunting in ants has already shown that experience with a nest of high- or low-quality influences the collective likelihood of accepting another nest: specifically, Healey and Pratt, 2008, showed that colonies housed in a low-quality nest for months accepted an average-quality new nest more readily than colonies housed for months in a high-quality nest. Furthermore, Stroeymeyt et al., 2011, showed that colonies housed in a high-quality nest developed an aversion towards low-quality available nests available in their environment, whereas colonies housed in a low-quality nest did not; and they developed an explanatory model similar to the alternative scenario I described in my first major comment. Please add these studies to the Introduction and discuss your results in light of these previous studies (individual mechanism unknown, but similar colony-level benefits as the one you refer to in the second paragraph of the Discussion).

Reviewer #2:

This is a very interesting study raising an original issue. The paper is well written. Unfortunately, while I appreciate the amount of work done in this study, in my opinion the paper suffers from several major issues that need to be addressed or at least discussed further. As it is the results do not yet support entirely the authors hypothesis.

- Experiment 1: one control group for each concentration (or a sub sample) would have been better than a single control as the results obtained might reflect the ant nutritional state (ants might be hungry when they received low concentration solutions while ants might be satiated when they received high concentration solutions). It is essential to show that the acceptance level would be the same between the 2nd and the 3rd visit for each concentration tested and not only for 0.5M. That would strengthen the results considerably and provide more conclusive evidences than experiment 2.

- Experiment 2: The results are not convincing enough.

First ants exposed to high molarity (1.5M) associated cues on their outwards journey showed a significantly higher number of pheromone depositions on their return journey after feeding on a 0.5M solution than ants confronted with low molarity scent (0.25M). Following the authors’ hypothesis and the results from experiment 1 the opposite is expected. Why this counterintuitive results? The author discussed the "experimental disruption effect" (Note 5 in supplementary material) but it can't explain why pheromones deposition is higher when ants are experiencing negative contrast effect.

Thus, the only result supporting the authors hypothesis is shown in Figure 3B. They show that ants exposed to 1.5M-associated cues showed significantly lower food acceptance towards the unscented 0.5M feeder than ants exposed to 0.25M-associated cues (P-Value 0.03). However the model run to compare the data does not include the factor "scent/molarity association". The authors used two different scents (each scent was associated with either the low or the high concentration). Even if the design is perfectly well balanced (well done!), the association ought to be added as a factor. When exploring the results (Excel file provided by the authors), ants exposed to high molarity associated cues facing a 0.5M solution showed an acceptance score of 0.65 (Lemon) or 0.74 (Rosemary) while ants confronted with low molarity scent facing a 0.5M solution showed an acceptance score of 0.92 (Lemon) or 0.77 (Rosemary). I understand that the authors want to focus on the difference between acceptance scores obtained when ants encountered high molarity associated cues or low molarity associated cues. However, while the difference is there, the absolute acceptance score for a solution associated with rosemary remain similar no matter the treatment while the score acceptance for Lemon varies. Thus, I think the authors should add the "scent/molarity association" to their model. In addition, as discussed in Note 5, in experiment 2 the acceptance score for a 0.5M solution (0.7) obtained when the ant encountered the high molarity associated cues (1.5M) is almost twice as high as the acceptance score (0.4) obtained when an ant was proposed a 0.5M after being offered 1.5 M. The authors discuss this results but did not test any of the hypothesis enunciated. They could for example measure the acceptance score for a 0.5M solution associated with Lemon or Rosemary (with an alternance between odors or no alternance between odors). I think it's crucial to understand why acceptance score vary so widely as it is the main behavioral response explored in this paper.

- Experiment 3: In ants and honeybees, It is well known that transfer rate decreased with increasing viscosity (Farina and Josens work) so it is not surprising that trophallaxis time is higher for high molarity solution. Thus, the authors cannot assert that trophallaxis time is due entirely to food quality. They need to discuss this effect or control for viscosity by adding different amounts of tylose (see Josens and Farina, 2001). Again here, the nutritional state could be an alternative hypothesis for the acceptance scores obtained (an ant that received a 1.5M solution via trophallaxis might be less hungry and less motivated than an ant that received a 0.16 solution via trophallaxis).

- I found the acceptance score (the main behavioral response used in this paper) a bit too qualitative. I don’t see a large difference between full acceptance (the ant remained in contact with the drop from the moment of contact and did not interrupt drinking within 3 seconds of initial contact) and partial acceptance (feeding was interrupted within 3 seconds after the first contact with the food source, but the ant still filled its crop). How does an interruption in feeding translate in acceptance? Is there any literature to support such evidence?

The authors then state “rejection was scored if the ant refused to feed at the sucrose solution and either returned to the nest immediately or failed to fill its crop within 10 minutes”.

=> so, from what I understand (and I might be wrong!), an ant half full or an ant that do not feed are given the same acceptance score (0) while ants that filled their crop completely but with or without interruption in the first 3 sec are given different scores (1 and 0.5 respectively). A better variable could be time spend feeding or actual volume ingested (see Mailleux et al., 2010). As the authors recorded their experiment with a camera, that could possibly be done.

Reviewer #3:

This is an interesting and generally well-designed and presented study on the application of prospect theory to ant foraging decisions. It convincingly shows (barring one issue described below) that an individual ant's assessment of a food source is strongly affected by the source's quality compared to a recently experienced reference. The main result is quite robust, and the authors performed a useful further experiment that at least partly discounts explanations based on sensory factors or nutritional state. I have a few critiques of the organization, presentation, and relationship to previous work.

1) The manuscript ignores highly relevant literature about the effects of recent female experience on mate choice. In essence, these studies report that a female becomes more (or less) choosy about a mate of given quality if she has prior experience of better (or worse mates). This literature includes studies both of vertebrates and invertebrates (a few examples: Collins, 1995; Dukas, 2005; Reid and Stamps, 1997; Wagner et al., 2001).

2) I am concerned about the behavioral definitions of food acceptance. The difference between full acceptance and partial acceptance seems slight and arbitrary. This may be a consequence of too brief a description.

3) As written, the results (especially for experiment 2) are very difficult to understand without first reading the Materials and methods. It would be better to preface each experiment's results with a brief description of its general design and rationale. This could most easily be done by transferring to the Results much of the experimental descriptions currently in the Materials and methods.

4) There is much information in the supplementary material that belongs in the main text. Notes 1 through 5 give important background and motivation for experimental designs, as well as discussion of key results. Some of the supplementary methods are in fact repeated in the main text, but there is much else that should be moved there. The supplementary material is best used for alternative analyses and graphical displays, details of statistical models, and very fine-grained method descriptions. The basic methods needed to understand the experiments well should be in the main text.

5) The legend to Figure 3 is hard to read. The opening sentence is simply too long and difficult to follow.

[Editors’ note: what now follows is the decision letter after the authors submitted for further consideration.]

Thank you for submitting your article "Positive and negative incentive contrasts lead to relative value perception in ants" for consideration by *eLife*. Your article has been reviewed by three peer reviewers, and the evaluation has been overseen by Diethard Tautz as the Senior and Reviewing Editor. The following individuals involved in review of your submission have agreed to reveal their identity: Nathalie Stroeymeyt (Reviewer #1); Stephen Pratt (Reviewer #2).

The reviewers have discussed the reviews with one another and the Reviewing Editor has drafted this decision to help you prepare a revised submission.

Summary:

The authors have adequately addressed the major comments. The clarity of the manuscript has also been much improved, as the authors moved crucial information from the Supplementary materials to the main text. However, some remaining problems have still been identified that need a further revision.

Essential revisions:

1) The paragraph discussing how social insects can be used as a model to study behavioural economics in humans (Discussion, tenth paragraph) is still a bit hand-wavy, and the additional sentences inserted by the authors are not so helpful (""Direct damage, or the scent of damaged or freshly killed ants, or predators, could, for example, be used to quantify the economic cost of death or massive physical damage. This would allow direct economic quantification of factors which cannot be inflicted on human subjects.") Given that the main text has significantly increased in length due to the addition of important information from the supplementary material, we recommend removing this paragraph.

2) The authors justify their choice of 3 seconds uninterrupted drinking as a cut-off between full and partial food acceptance based on their 'extensive experience' with the ants' behaviour, and provide two videos to illustrate the difference between the two scores. However, the videos are not included – they need to be made accessible. Furthermore, the justification of this apparently arbitrary score remains unclear. Could one simply have counted the number of interruptions until the ants' crop was filled and used this as a continuous measure instead? The authors also say that "a 2-second or 4-second window would probably have given the same results", but they have not actually checked. Repeating the analyses using these thresholds and finding that the results are indeed qualitatively the same would go a long way towards justifying the authors' choice.

3) In experiment 2 the authors state: "40 ants were induced to expect a high reward, and 33 to expect a low reward." (while in reality it is 42 and 32 respectively). Looking at the table uploaded on dryad, in total 74 ants were trained and 2 had missing data for the test (visit 9). Thus, 72 observations should have been included in the analysis. However, the number of observations indicated in the model output is 73 (subsection “Statistical analysis”).

As the main results of experiment 2 hold on this particular statistical test (P=0.03), it should be run properly.

4) I reiterate my comment: in the Cumulative Link Mixed Model it is indicated that two measures were performed on the same ant (visit 1 and visit 2) but the factor, "visitnr" is treated as a fixed factor and the ant ID is not included in the model.

5) Throughout the statistics, the authors often added the random factors as (1 | Colony) + (1 | AntID) but some other time they added them as (1 | Colony/AntID) i.e. (AntID nested in colony). A justification is required why this has been done.

---

## [Author Response]

Reviewer #1:[…] 1) The results from experiment 1 and 2 could be explained by an alternative hypothesis in which each ant has a fixed, experience-independent threshold for food acceptance. (This alternative hypothesis is inspired from the following work on house-hunting in ants: Robinson et al., 2009, Robinson et al., 2011, Stroeymeyt et al., 2011).

Subsection “Non-random selection of individuals with different acceptance thresholds”. This alternative hypothesis was added to the introduction of experiment 2 and discussed in light of our results (subsection “Experiment 2 – Methods”). Our data and experimental design allow us to rule out this hypothesis (see detailed response below).

“Non-random selection of individuals with different acceptance thresholds: different individuals of the same colony may have different acceptance thresholds. […] However, we can exclude this possibility, as the proportion of ants not completing training was uniformly low and did not vary with treatment (see Supplementary file 1).”

“Similar results in house-hunting ants were explained by a simple threshold rule (Stroeymeyt et al., 2011; Robinson et al., 2009, 2011) which suggests that individuals have different acceptance thresholds and ants with lower thresholds accept lower quality options. […] It is thus unlikely that a simple threshold rule led to the results shown in experiment 1 (Figure 2 and 3).”

In experiment 1, the authors explain that they allow 4 ants on the drawbridge, and wait until the first ant finds the food. That ant becomes the focal experimental ant and her behavior is monitored during that visit plus two subsequent trips, then is discarded from the colony. However, no information is given regarding whether any ant had to be discarded from the experiment because they did not perform three trips. One could imagine a scenario in which ants are more likely to lose motivation to forage when the food source they found is of lower quality than their internal, fixed acceptance threshold. Under that scenario, when trained with a food source of very high quality, all or most ants would keep motivation to forage, whereas when trained with a food source of low quality, only ants with a low acceptance threshold would keep motivation to forage. Thus during the third, test visit, the populations of tested foragers would not be equivalent between treatments but would show a sampling bias: in the first treatment (high quality training food), tested foragers would present a broad variation of acceptance threshold and would thus be more likely to reject an average-quality food source during the third visit, whereas in the second treatment (low quality training food), tested foragers would only include ants with low acceptance threshold, which would thus all accept an average-quality food source. In order to reject this scenario, it is necessary that the authors present data on the proportion of focal ants that did not do three trips. If that proportion does not increase with decreasing quality of the training food, then the alternative hypothesis I presented can be safely rejected, which would provide more support for the authors' interpretation.

This is a very insightful hypothesis, and one which had not occurred to us. Indeed, the reviewer is completely correct that this could present a critical flaw in our experiment. However, luckily, the proportion of ants, which did not participate in all three training visits did not increase with decreasing quality of the training food. Supplementary file 1 shows the complete data of ants, which completed all three visits, and ants, which cancelled foraging before training and data collection was complete.

Two columns (Number of ants excluded and proportion finished) were added to Supplementary file 1 and a mean of all successfully finished trials added to the Materials and methods section of experiment 1: “97% of ants successfully finished the training procedure and participated in the test visit (third visit).”.

Similarly, in experiment 2, it is possible that ants with a high acceptance threshold make a U-turn and return to the nest when reaching the scented portion of the bridge when it carries the smell associated with the low-quality food, while all ants may cross the bridge and reach the food when the bridge carries the smell associated with the high-quality food. Under that scenario, the two test populations would again not be equivalent, and the results would be explained by self-organized sampling bias rather than by a cognitive expectation effect. I would thus like the author to include data on the proportion of ants that performed U-turns when reaching the scented portion of the bridge in the two treatments. If there is no difference between the two treatments, then the alternative hypothesis can again be safely rejected and this would provide more support for the authors' interpretation.

In experiment 2, there were no ants which returned to the nest on the test visit. All ants tasted the food at the end of the runway regardless of motivation when reaching the runway scent, and food acceptance scores were noted for each ant which completed training. Furthermore, all ants received the same training procedure and had to taste each sucrose droplet during training. Motivation generally dropped when confronted with a scent which was associated to a low molarity and ants performed U-turns during later training visits. However, the sucrose droplet had to be tasted on each visit in order to complete training. Thus, during the test visit, we also get food acceptance scores from ants which performed more U-turns and showed lower motivation compared to others.

However, even if a filter was in place and we were only testing the high motivation ants (which seems unlikely, given the information in Table 1), this would be the case for both treatments, because treatments were identical until the final test visit, in which only the odour of the runway changed. While it is in principle possible that on the test visit ‘high threshold’ ants would be filtered out, no ants which completed training returned to the nest during the testing trial and did not complete testing.

Finally, I am not convinced that the results from experiment 3 cannot be solely explained by the different crop loads of the receiver ants in the three treatments: as the receiver ants receive more food from donor ant when the original food source is of higher quality, and as they were not given any opportunity to unload before being confronted with the test food, then one would predict them to drink less during the test (and so to have a lower food acceptance score) simply because they have less space in their crop. I am not sure that the statistical analyses presented by the authors are enough to demonstrate an effect beyond this simple mechanical effect. Instead of Figure S6.3, can the author present a scatterplot of mean acceptance score as a function of trophallaxis time for each of the three treatments and elaborate on the interpretation of the statistical analyses presented in the Supplementary material? Given that the main effect 'Reference Molarity' is non-significant and that the sign of the estimate (+) is opposite to that expected based on the author's hypothesis (higher reference molarity should result in lower food acceptance), it seems that the conclusions drawn from experiment 3 hold may not be supported by the data. Please elaborate and modify the text if relevant.

This is a reasonable point. Firstly, it may not be that ants which showed higher trophallaxis times also ingest more sucrose solution during trophallaxis. Higher molarities may lead to longer trophallaxis times simply due to the higher viscosity of the solution while the amount of transferred food stays the same for all three molarities (Josens, Farina, and Roces, 1998; Tezze and Farina, 1999). Even if more food was transferred, the food acceptance scores are a measure of the first assessment of ants at a food source, not the ingested volume. Thus, while some ants may have had less space in their crop left, this may not necessarily affect the food acceptance score, while it is very likely to affect ingested volume after trophallaxis.

Additionally, if longer trophallaxis times lead to more ingested sucrose solution, it is also more likely that a higher amount of information about the past food quality is transferred. Thus, longer trophallaxis times may also be expected to lead to stronger contrast effects due to more information being transferred through trophallaxis.

With all these caveats, the comments of the reviewer are still valid, and so we carried out the requested analyses. The data shows clear effects of both trophallaxis time and reference solution on the food acceptance towards 0.5M sucrose. Even at high trophallaxis times, ants with a 0.16M reference showed no low food acceptance scores, unlike ants with high reference solution after long trophallaxis times.

The fixed effects of the model indeed seemed a bit odd and as the data seemed to show a very different pattern, so we ran the model again with scaled continuous variables as suggested for GLMMs. While the effect of the interaction stayed the same in the updated model, we also see a clearer effect of the fixed effects, confirming that both trophallaxis time and reference molarity affect the food acceptance at the post-shift solution – as shown in Author response image 1 and 2. Thank you once again for the suggestion!

**Author response image 2. respfig2:** 

We now add this analysis and figures to Figure 5—source data 1, and discuss this hypothesis briefly in the Discussion: “If, however, more sucrose solution was transferred between the returning forager and the recruit at longer trophallaxis times, it is likely that information input increases and food acceptance decreases. […] Even at high trophallaxis times, ants with a 0.16M reference showed no low food acceptance scores, unlike ants with high reference solution after long trophallaxis times.”

2) When reading the main text, I had trouble evaluating the solidity of the authors' claim because of the lack of key information which is only provided in the Supplementary material. In particular, please provide the following information in the main text:- definition of the food acceptance score

This was added to the Materials and methods section of experiment. For clarity, we will also add videos for the scores 1 and 0.5 as additional supplements.

“Food acceptance scores as a measure of perceived value were noted for each ant and visit as follows: As the ant drank at the droplet it was given one of three food acceptance scores. […] Lastly, rejection (0) was scored if the ant refused to feed at the sucrose solution and either returned to the nest immediately or failed to fill its crop within 10 minutes.”

- information that each ant was uniquely identified owing to the design of the experiment (each experimental ant monitored fully then discarded from the colony before switching to the next ant)

This was added to the Materials and methods section: “Each tested ant was observed until all experimental runs were finished and then discarded from the colony before switching to the next ant.”

- more detail on the alternative scenarios at the beginning of experiment 2

This was added to the introductory part of experiment 2 (subsection “Experiment 2 – ruling out alternative explanations using scent training”).

- more detail on the experimental ant selection procedure, to show that your experimental design does not introduce any a priori selection bias between treatments

This was added to the ant selection and monitoring section: “This procedure may select for the more active foragers, but does not introduce any selection bias between treatments.”

3) Previous work on house-hunting in ants has already shown that experience with a nest of high- or low-quality influences the collective likelihood of accepting another nest: specifically, Healey and Pratt, 2008, showed that colonies housed in a low-quality nest for months accepted an average-quality new nest more readily than colonies housed for months in a high-quality nest. Furthermore, Stroeymeyt et al., 2011, showed that colonies housed in a high-quality nest developed an aversion towards low-quality available nests available in their environment, whereas colonies housed in a low-quality nest did not; and they developed an explanatory model similar to the alternative scenario I described in my first major comment. Please add these studies to the Introduction and discuss your results in light of these previous studies (individual mechanism unknown, but similar colony-level benefits as the one you refer to in the second paragraph of the Discussion).

This was added to the general Introduction, introduction of experiment 2 (subsection “Non-random selection of individuals with different acceptance thresholds”), and the Discussion of the main manuscript.

“Healey and Pratt, 2008, showed that colonies of the house-hunting ant species *Temnothorax curvispinosus* move into a nest of mediocre quality faster when they were previously housed in a high-quality nest compared to colonies which were previously housed in a poor-quality nest (Healey and Pratt, 2008). In contrast, Stroeymeyt et al., 2011, showed that colonies of *Temnothorax albipennis* developed an aversion towards mediocre-quality nests available in their environment when they were housed in a high-quality nest, whereas colonies housed in a low-quality nest did not, and thus show an experience-dependent flexibility in nest choice (Stroeymeyt et al., 2011).”

“Non-random selection of individuals with different acceptance thresholds: different individuals of the same colony may have different acceptance thresholds. […] However, we can exclude this possibility, as the proportion of ants not completing training was uniformly low and did not vary with treatment (see Supplementary file 1).”

“Similar results in house-hunting ants were explained by a simple threshold rule (Stroeymeyt et al., 2011; Robinson et al., 2009, 2011) which suggests that individuals have different acceptance thresholds and ants with lower thresholds accept lower quality options. […] It is thus unlikely that a simple threshold rule led to the results shown in experiment 1 (Figure 2 and 3).”

“For example, colonies of house-hunting ants developed an aversion towards mediocre nests when housed in high-quality nests, but not when they were housed in low-quality nests. […] However, while in house-hunting the reference resource is directly experienced by scouts only, we demonstrate that information brought back to the nest can set a reference point for ants which have not directly experienced the resource in situ.”Reviewer #2:This is a very interesting study raising an original issue. The paper is well written. Unfortunately, while I appreciate the amount of work done in this study, in my opinion the paper suffers from several major issues that need to be addressed or at least discussed further. As it is the results do not yet support entirely the authors hypothesis.- Experiment 1: one control group for each concentration (or a sub sample) would have been better than a single control as the results obtained might reflect the ant nutritional state (ants might be hungry when they received low concentration solutions while ants might be satiated when they received high concentration solutions). It is essential to show that the acceptance level would be the same between the 2nd and the 3rd visit for each concentration tested and not only for 0.5M. That would strengthen the results considerably and provide more conclusive evidences than experiment 2.

As all tested colonies were starved 4 days prior to experiments and foragers collect food for the whole colony, it is very unlikely that the single forager or even the whole colony was satiated from a maximum of 30 crop loads during a testing day. We can provide data for a 1M food source from our lab at which food acceptance on the second visit is at 0.88, and for the third visit at 0.87 (n=353 for both visits), suggesting no change in food acceptance from the second to third visit. While adding this information would indeed add elegance, this would require an effective doubling of the number of trials. As experiment 1 required over two months to perform, this seems to us excessive.

Finally, this issue was addressed with experiment 2 where all ants received the same training procedure and were thus supposed to be at a similar hunger level. Collecting more data on other molarities would also not rule out other explanations as experiment 2 does.

- Experiment 2: The results are not convincing enough.First ants exposed to high molarity (1.5M) associated cues on their outwards journey showed a significantly higher number of pheromone depositions on their return journey after feeding on a 0.5M solution than ants confronted with low molarity scent (0.25M). Following the authors’ hypothesis and the results from experiment 1 the opposite is expected. Why this counterintuitive results? The author discussed the "experimental disruption effect" (Note 5 in supplementary material) but it can't explain why pheromones deposition is higher when ants are experiencing negative contrast effect.

This is indeed an important point, and we thank the reviewer for raising it and asking us to think carefully about it again. After careful consideration of the methods and data, we believe we have uncovered the reason for this pattern in the pheromone deposition data:

The scented paper overlays were replaced with unscented ones. However, we did not replace the runways on which the paper overlays were placed. It is thus possible that small portions of the previous odour was still present. The ants were thus likely reacting to the remaining odour as associated in the training visit (that is, depositing more pheromone for the ‘good’ smell and less for the ‘poor’ smell), overlaid over a major change and reduction in deposition due to a disruption effect.

Comparing the pheromone data with data collected in a separate experiment currently in preparation supports this hypothesis. In this experiment the scented overlays were *not* replaced after the last visit. We see the same pattern of pheromone deposition in the current experiment. Ants were confronted with scented overlays and scented sucrose solution also in the test visit (0.387M post-shift). We show this data in Author response image 3: panel A shows the number of pheromone depositions during 8 training visits with 1.5M as high molarity and 0.1M as low molarity. Panel B shows pheromone depositions on the 9^th^ visit with 0.387M scented sucrose solution as post-shift reward and scented paper overlays on the way back to the nest. Panel C shows the data from the 9^th^ visit of the current experiment 2, for comparison. Note that B and C are very similar. However, since both training molarities and test molarities differed between the experiments, and the data was collected in different years and mostly with different colonies, direct comparisons should be made with caution.

We now discuss the reason for the unexpected pheromone deposition pattern: “Furthermore, since only the scented paper overlays were replaced by unscented ones, but not the runways themselves, it is possible that small portions of the odours were still present, driving the ants to deposit pheromone according to the remaining odours, with higher depositions rates for the high-quality associated odour. […] This would explain why pheromone depositions were higher for ants returning to the nest from a high molarity scent than in ants returning from a low molarity scent.”

**Author response image 3. respfig3:** 

Thus, the only result supporting the authors hypothesis is shown in Figure 3B. They show that ants exposed to 1.5M-associated cues showed significantly lower food acceptance towards the unscented 0.5M feeder than ants exposed to 0.25M-associated cues (P-Value 0.03). However the model run to compare the data does not include the factor "scent/molarity association". The authors used two different scents (each scent was associated with either the low or the high concentration). Even if the design is perfectly well balanced (well done!), the association ought to be added as a factor. When exploring the results (Excel file provided by the authors), ants exposed to high molarity associated cues facing a 0.5M solution showed an acceptance score of 0.65 (Lemon) or 0.74 (Rosemary) while ants confronted with low molarity scent facing a 0.5M solution showed an acceptance score of 0.92 (Lemon) or 0.77 (Rosemary). I understand that the authors want to focus on the difference between acceptance scores obtained when ants encountered high molarity associated cues or low molarity associated cues. However, while the difference is there, the absolute acceptance score for a solution associated with rosemary remain similar no matter the treatment while the score acceptance for Lemon varies. Thus, I think the authors should add the "scent/molarity association" to their model. In addition, as discussed in Note 5, in experiment 2 the acceptance score for a 0.5M solution (0.7) obtained when the ant encountered the high molarity associated cues (1.5M) is almost twice as high as the acceptance score (0.4) obtained when an ant was proposed a 0.5M after being offered 1.5 M. The authors discuss this results but did not test any of the hypothesis enunciated. They could for example measure the acceptance score for a 0.5M solution associated with Lemon or Rosemary (with an alternance between odors or no alternance between odors). I think it's crucial to understand why acceptance score vary so widely as it is the main behavioral response explored in this paper.

This is a fair point. The scent/molarity association was added to the model as suggested by the reviewer. There is still a significant effect of expected molarity, but however no significant effect of the odour used (see model summary below).

Cumulative Link Mixed Model fitted with the Laplace approximation

formula: newFoodAcceptability ~ Scent.Molarity * HighLowMolarityscent + (1 | Colony)

data: visit9FA

link threshold nobs logLik AIC niter max.grad cond.H

logit flexible 72 -57.86 127.72 281(385) 1.62e-05 3.8e+04

Random effects:

Groups Name Variance Std.Dev.

Colony (Intercept) 0.003578 0.05982

Number of groups: Colony 6

Coefficients:

Estimate Std. Error z value Pr(>|z|)

Scent.MolarityRosemary 0.3924 0.6280 0.625 0.5321

HighLowMolarityscentLow 1.9828 0.9007 2.201 0.0277 *

Scent.MolarityRosemary:HighLowMolarityscentLow -1.5018 1.0953 -1.371 0.1703

---

Signif. codes: 0 ‘***’ 0.001 ‘**’ 0.01 ‘*’ 0.05 ‘.’ 0.1 ‘’ 1

Threshold coefficients:

Estimate Std. Error z value

0|0.5 -2.3120 0.6309 -3.665

0.5|1 0.3579 0.4580 0.781

**Author response image 4. respfig4:** 

We do not have data on acceptance scores for scented 0.5M sucrose. However, we can provide acceptance data of ants feeding on a 1M sucrose solution associated to Lemon over successive visits (visit1: 0.99, visit 2: 0.89, visit 3: 0.89) or Rosemary scent (visit 1: 0.97, visit 2: 0.86, visit 3: 0.84), showing that acceptance scores vary from the first to second visits but then remain rather constant when confronted with the same odour and molarity. This data was collected by Oberhauser and Czaczkes and partly published in Biology Letters (Oberhauser and Czaczkes, 2018).

Understanding why acceptance scores vary so widely is a huge undertaking, and is effectively attempting to answer the question “what drives relative value perception?” Our manuscript covers one important driver of relative value perception: expectations. However, fully answering such a broad question could fill an entire career. We hope we have begun to make good inroads into answering this question, as have others before us, but we are still far from a comprehensive answer.

- Experiment 3: In ants and honeybees, It is well known that transfer rate decreased with increasing viscosity (Farina and Josens work) so it is not surprising that trophallaxis time is higher for high molarity solution. Thus, the authors cannot assert that trophallaxis time is due entirely to food quality. They need to discuss this effect or control for viscosity by adding different amounts of tylose (see Josens and Farina, 2001). Again here, the nutritional state could be an alternative hypothesis for the acceptance scores obtained (an ant that received a 1.5M solution via trophallaxis might be less hungry and less motivated than an ant that received a 0.16 solution via trophallaxis).

The fact that higher viscosities also lead to longer trophallaxis times has been made clearer in the manuscript. However, while this effect is well known, our attention is on the effect of the quality of reference solution on the food acceptance when confronted with medium quality sucrose, not on the trophallaxis times. We completely agree that experiment 3 cannot rule out alternative explanations such as different hunger levels. Only experiment 2 was designed to rule these alternatives out.

“Even though higher trophallaxis times led to lower acceptance scores and trophallaxis times were higher at high reference molarities, this does not necessarily imply that ants ingested more sucrose at higher references and were thus less hungry or motivated. […] Even at high trophallaxis times, ants with a 0.16M reference showed no low food acceptance scores, unlike ants with high reference solution after long trophallaxis times.”

- I found the acceptance score (the main behavioral response used in this paper) a bit too qualitative. I don’t see a large difference between full acceptance (the ant remained in contact with the drop from the moment of contact and did not interrupt drinking within 3 seconds of initial contact) and partial acceptance (feeding was interrupted within 3 seconds after the first contact with the food source, but the ant still filled its crop). How does an interruption in feeding translate in acceptance? Is there any literature to support such evidence?

Videos have now been added (Video 1 and Video 2) to illustrate the foraging behavior for acceptance scores 1 and 0.5. Ants which displayed score 1 were generally very calm while feeding, while in contrast ants which displayed score 0.5 tend to show multiple feeding interruptions and show ‘impatience’ while feeding.

The authors then state “rejection was scored if the ant refused to feed at the sucrose solution and either returned to the nest immediately or failed to fill its crop within 10 minutes”.=> so, from what I understand (and I might be wrong!), an ant half full or an ant that do not feed are given the same acceptance score (0) while ants that filled their crop completely but with or without interruption in the first 3 sec are given different scores (1 and 0.5 respectively). A better variable could be time spend feeding or actual volume ingested (see Mailleux et al., 2010). As the authors recorded their experiment with a camera, that could possibly be done.

This is a reasonable comment. Unfortunately, time spent feeding is apparently not a very good variable here, because higher viscosity leads to higher drinking times in ants (Josens, Farina, and Roces, 1998) and increasing sucrose molarities lead to higher viscosity of the solutions. While we attempted to extract a measure of ingested volume from the videos, it was not possible to do this reliably due to limitations in video resolution.

We hope that the videos now make the difference in food acceptance scores clearer. We think the scores delineate biologically reasonable points in terms of acceptance: a 0 score almost always represents ants which originally showed score 0.5, but then cancelled the foraging trip without filling their crop. It is worth nothing that scores of 0 were rather rare.

Finally, we reiterate that the scoring was performed blind to treatment, which suggests that this is a reliable indicator that acceptance correlates with value.

Reviewer #3:This is an interesting and generally well-designed and presented study on the application of prospect theory to ant foraging decisions. It convincingly shows (barring one issue described below) that an individual ant's assessment of a food source is strongly affected by the source's quality compared to a recently experienced reference. The main result is quite robust, and the authors performed a useful further experiment that at least partly discounts explanations based on sensory factors or nutritional state. I have a few critiques of the organization, presentation, and relationship to previous work.1) The manuscript ignores highly relevant literature about the effects of recent female experience on mate choice. In essence, these studies report that a female becomes more (or less) choosy about a mate of given quality if she has prior experience of better (or worse mates). This literature includes studies both of vertebrates and invertebrates (a few examples: Collins, 1995; Dukas, 2005; Reid and Stamps, 1997; Wagner et al., 2001).

This was added to the Introduction of the main manuscript. Many thanks for pointing our attention towards these relevant studies.

“Incentive contrasts have also been demonstrated for rewards other than food. Females become more (or less) likely to accept a mate of given quality if they have prior experience of better (or worse) mates. Such mate quality contrast effects are reported in both vertebrates (Collins, 1995) and invertebrates (Dukas, 2005; Reid and Stamps, 1997; Wagner, Smeds, and Wiegmann 2001).”

2) I am concerned about the behavioral definitions of food acceptance. The difference between full acceptance and partial acceptance seems slight and arbitrary. This may be a consequence of too brief a description.

Videos and a more detailed description were added to the main manuscript. See also our response to points by reviewer 2.

“Food acceptance scores as a measure of perceived value were noted for each ant and visit as follows: when the ant drank at the droplet it was given one of three food acceptance scores. […] Lastly, rejection (0) was scored if the ant refused to feed at the sucrose solution and either returned to the nest immediately or failed to fill its crop within 10 minutes.”

3) As written, the results (especially for experiment 2) are very difficult to understand without first reading the Materials and methods. It would be better to preface each experiment's results with a brief description of its general design and rationale. This could most easily be done by transferring to the Results much of the experimental descriptions currently in the Materials and methods.

We agree, and have completely restructured the manuscript to increase clarity. We now describe the methods and results of each experiment together continuing to the next experiment.

4) There is much information in the supplementary material that belongs in the main text. Notes 1 through 5 give important background and motivation for experimental designs, as well as discussion of key results. Some of the supplementary methods are in fact repeated in the main text, but there is much else that should be moved there. The supplementary material is best used for alternative analyses and graphical displays, details of statistical models, and very fine-grained method descriptions. The basic methods needed to understand the experiments well should be in the main text.

The supplementary notes 1 to 5 were added to the main manuscript. Many methodological details have also now been moved to the main manuscript.

5) The legend to Figure 3 is hard to read. The opening sentence is simply too long and difficult to follow.

Agreed. This has now been reworded.

[Editors’ note: what now follows is the decision letter after the authors submitted for further consideration.]Essential revisions:1) The paragraph discussing how social insects can be used as a model to study behavioural economics in humans (Discussion, tenth paragraph) is still a bit hand-wavy, and the additional sentences inserted by the authors are not so helpful (""Direct damage, or the scent of damaged or freshly killed ants, or predators, could, for example, be used to quantify the economic cost of death or massive physical damage. This would allow direct economic quantification of factors which cannot be inflicted on human subjects.") Given that the main text has significantly increased in length due to the addition of important information from the supplementary material, we recommend removing this paragraph.

These sentences have now been removed from the manuscript.

2) The authors justify their choice of 3 seconds uninterrupted drinking as a cut-off between full and partial food acceptance based on their 'extensive experience' with the ants' behaviour, and provide two videos to illustrate the difference between the two scores. However, the videos are not included – they need to be made accessible. Furthermore, the justification of this apparently arbitrary score remains unclear. Could one simply have counted the number of interruptions until the ants' crop was filled and used this as a continuous measure instead? The authors also say that "a 2-second or 4-second window would probably have given the same results", but they have not actually checked. Repeating the analyses using these thresholds and finding that the results are indeed qualitatively the same would go a long way towards justifying the authors' choice.

The videos had been made accessible via the offered Dryad link. We apologize for not making this clearer and easier to access them. However, the videos have now been updated directly through the submission form as video 1 (showing food acceptance score 1) and video 2 (showing food acceptance score 0.5).

The results of experiment 2 have been re-analysed from the videos in order to get data on the number of drinking interruptions until the crop is filled, and to be able to compare the 2, 3 and 4 second windows of food acceptance. This also represents an independent blind assessment of the ants’ behaviour, as the analysis was performed by a naïve analyser unfamiliar with the study. Due to the very large amount of videos and time limitations, it was not feasible for us to re-analyse the data of all three experiments from the videos. We thus chose to focus on experiment 2, as it was the focus of much of reviewer 2’s concerns. Unfortunately, two of the videos were missing, resulting in a slightly lower sample size (68 instead of 70).

We found a slightly stronger significant effect of the expected molarity based on the runway scent in the 3- and 4-second windows of food acceptance than that presented in the main manuscript (3 seconds CLMM: estimate=1.42, z=2.35, p=0.019, 4 seconds CLMM: estimate=1.27, z=2.28, p=0.023). While the same trend is maintained for the 2-second window, the groups were not significantly different using this definition (estimate=0.88, z=1.32, p=0.19). However, pleasingly, the number of food interruptions until the crop was filled was significantly lower when ants expected to find low molarity food, but got medium quality (positively surprised) compared to when they expected high molarity food but got medium (disappointed) (GLMM: estimate=-0.35, z=-2.74, p=0.006, see Author response image 5).

We furthermore analysed first interruption times of the tested ants on the last (9^th^) visit. Surprisingly, we found no significant effect of ant’s expectations on the first interruption time (GLMM: estimate=-0.34, z=-1.07, p=0.28), although a visual inspection of the data suggested it should be there (see Author response image 6). There was, however, a highly significant effect of the used odour on the first interruption time (GLMM: estimate=1.19, z=3.7, p< 0.001). Author response image 6 shows that first interruption times were higher in ants which expected to receive low molarity food compared to ants which expected high molarity food. Since we found a significant effect of odour, we split the data according to odour presented on the 9^th^ visit and analysed the subsets once again, which nicely explained the results: Author response image 7 clearly shows a strong effect of expectations on first interruption times when the lemon odour was used (GLMM: estimate=1.74, z=3.92, p<0.001), but a weaker effect, although still in the same direction when rosemary odour was used (GLMM: estimate=0.63, z=1.44, p=0.15).

Unfortunately, we cannot explain the differences between lemon and rosemary odours in these experiments. Lemon and rosemary odours are used in our conditioning experiments, because ants did not show a preference for any of the odours in previous experiments. It is possible that ants are more able to associate food quality (either high or low) with lemon odour than rosemary odour.

We now briefly mention the blind reanalysis of acceptance scores using the 3 second window, and the analysis of interruption times, in the main text (subsection “Experiment 2 – Results”). We hope that the data collected from the re-analysis of the second experiment, as well as the videos showing food acceptance scores 1 and 0.5, now sufficiently justify the use of our 3-second food acceptance scores as a measure of the attractiveness of a food source in these experiments.

Graph and statistical analysis for the number of food interruptions:

**Author response image 5. respfig5:** Number of food interruptions on the last (9^th^) visit depending on the ant’s expectations until the crop was filled.

## Generalized linear mixed model fit by maximum likelihood (Laplace

## Approximation) [glmerMod]

## Family: poisson (log)

## Formula: Pauses ~ HighLow + Scent + (1 | Colony)

## Data: alldat

## Control: glmerControl(optCtrl = list(maxfun = 10000))

##

## AIC BIC logLik deviance df.resid

## 321.6 330.5 -156.8 313.6 64

##

## Scaled residuals:

## Min 1Q Median 3Q Max

## -2.1614 -0.8313 -0.3108 0.6991 5.2410

##

## Random effects:

## Groups Name Variance Std.Dev.

## Colony (Intercept) 4.975e-17 7.053e-09

## Number of obs: 68, groups: Colony, 6

##

## Fixed effects:

## Estimate Std. Error z value Pr(>|z|)

## (Intercept) 1.3896 0.1055 13.167 < 2e-16 ***

## HighLowLow -0.3551 0.1295 -2.741 0.00613 **

## ScentRosemary 0.1520 0.1273 1.194 0.23248

## ---

## Signif. codes: 0 '***' 0.001 '**' 0.01 '*' 0.05 '.' 0.1 ' ' 1

##

## Correlation of Fixed Effects:

## (Intr) HghLwL

## HighLowLow -0.391

## ScentRosmry -0.665 -0.095

Graph and statistical analysis for the first interruption times including the whole data of experiment 2:

**Author response image 6. respfig6:** First interruption times [seconds] on the last (9^th^) visit depending on ant’s expectations for the complete data.

## Generalized linear mixed model fit by maximum likelihood (Laplace

## Approximation) [glmerMod]

## Family: poisson (log)

## Formula:

## as.integer(FirstInterruption) ~ HighLow + Scent + (1 | Colony/AntID)

## Data: alldat

## Control: glmerControl(optCtrl = list(maxfun = 10000))

##

## AIC BIC logLik deviance df.resid

## 568.8 579.9 -279.4 558.8 63

##

## Scaled residuals:

## Min 1Q Median 3Q Max

## -2.46587 -0.20136 -0.01499 0.07549 1.20787

##

## Random effects:

## Groups Name Variance Std.Dev.

## AntID:Colony (Intercept) 2.616e+00 1.617e+00

## Colony (Intercept) 2.071e-10 1.439e-05

## Number of obs: 68, groups: AntID:Colony, 67; Colony, 6

##

## Fixed effects:

## Estimate Std. Error z value Pr(>|z|)

## (Intercept) 1.5870 0.3531 4.495 6.95e-06 ***

## HighLowLow -0.3400 0.3175 -1.071 0.284268

## ScentRosemary 1.1931 0.3218 3.708 0.000209 ***

## ---

## Signif. codes: 0 '***' 0.001 '**' 0.01 '*' 0.05 '.' 0.1 ' ' 1

##

## Correlation of Fixed Effects:

## (Intr) HghLwL

## HighLowLow -0.663

## ScentRosmry -0.737 0.498

Graph and statistical analysis for the first interruption times for data subsets of lemon and rosemary odours:

**Author response image 7. respfig7:** First interruption times [seconds] on the last (9^th^) visit depending on ant’s expectations. Data for ants being confronted with lemon and rosemary odours on the runways were split.

GLMM for the Lemon Odour:

## Generalized linear mixed model fit by maximum likelihood (Laplace

## Approximation) [glmerMod]

## Family: poisson (log)

## Formula: as.integer(FirstInterruption) ~ HighLow + (1 | Colony/AntID)

## Data: lemonsubset

## Control: glmerControl(optCtrl = list(maxfun = 10000))

##

## AIC BIC logLik deviance df.resid

## 239.5 245.1 -115.7 231.5 26

##

s## Scaled residual:

## Min 1Q Median 3Q Max

## -1.06113 -0.30094 0.01654 0.08866 0.26499

##

## Random effects:

## Groups Name Variance Std.Dev.

## AntID:Colony (Intercept) 1.276e+00 1.129e+00

## Colony (Intercept) 6.094e-10 2.469e-05

## Number of obs: 30, groups: AntID:Colony, 30; Colony, 6

##

## Fixed effects:

## Estimate Std. Error z value Pr(>|z|)

## (Intercept) 1.5550 0.2940 5.289 1.23e-07 ***

## HighLowLow 1.7416 0.4435 3.927 8.60e-05 ***

## ---

## Signif. codes: 0 '***' 0.001 '**' 0.01 '*' 0.05 '.' 0.1 ' ' 1

##

## Correlation of Fixed Effects:

## (Intr)

## HighLowLow -0.658

GLMM for the Rosemary Odour:

## Generalized linear mixed model fit by maximum likelihood (Laplace

## Approximation) [glmerMod]

## Family: poisson (log)

## Formula: as.integer(FirstInterruption) ~ HighLow + (1 | Colony/AntID)

## Data: rosemarysubset

## Control: glmerControl(optCtrl = list(maxfun = 10000))

##

## AIC BIC logLik deviance df.resid

## 292.9 299.4 -142.4 284.9 34

##

## Scaled residuals:

## Min 1Q Median 3Q Max

## -1.02190 -0.31663 -0.03179 0.09659 0.19368

##

## Random effects:

## Groups Name Variance Std.Dev.

## AntID:Colony (Intercept) 1.606e+00 1.267e+00

## Colony (Intercept) 5.220e-10 2.285e-05

## Number of obs: 38, groups: AntID:Colony, 38; Colony, 6

##

## Fixed effects:

## Estimate Std. Error z value Pr(>|z|)

## (Intercept) 1.7210 0.3163 5.441 5.31e-08 ***

## HighLowLow 0.6292 0.4360 1.443 0.149

## ---

## Signif. codes: 0 '***' 0.001 '**' 0.01 '*' 0.05 '.' 0.1 ' ' 1

##

## Correlation of Fixed Effects:

## (Intr)

## HighLowLow -0.721

3) In experiment 2 the authors state: "40 ants were induced to expect a high reward, and 33 to expect a low reward." (while in reality it is 42 and 32 respectively). Looking at the table uploaded on dryad, in total 74 ants were trained and 2 had missing data for the test (visit 9). Thus, 72 observations should have been included in the analysis. However, the number of observations indicated in the model output is 73 (subsection “Statistical analysis”).As the main results of experiment 2 hold on this particular statistical test (P=0.03), it should be run properly.

Many thanks for bringing these two mistakes to our attention. The data was re-checked and it seems that two doubles were included in the data file, probably when the data was re-analysed through the videos. We thus re-ran all statistics for this experiment and corrected the mistakes in the manuscript, Figure 4—source data 1-6 and the Raw Data File. The results from experiment 2 remain unchanged, as do their significance (p = 0.04).

4) I reiterate my comment: in the Cumulative Link Mixed Model it is indicated that two measures were performed on the same ant (visit 1 and visit 2) but the factor, "visitnr" is treated as a fixed factor and the ant ID is not included in the model.

AntID was now included in the Cumulative Link Mixed Model (see Figure 2 – Source Data 1). The results of the model remain largely unchanged.

5) Throughout the statistics, the authors often added the random factors as (1 | Colony) + (1 | AntID) but some other time they added them as (1 | Colony/AntID) i.e. (AntID nested in colony). A justification is required why this has been done.

This had been done in order to achieve a better model fit. This change did however not affect the model fit and was now, in the process of re-analysing the whole experiment 2 section, changed to the correct version (AntID nested in colony) throughout the whole dataset.

The same random effects were added in the GLMMs for pheromone depositions of the last visits of experiment 2 in order to correct for overdispersion and obtain a better fit for the models. This was also done in the GLMM for the inbound pheromone depositions on visit 3 of experiment 1, also to correct for overdispersion.